# Neurokinin-3 receptor activation selectively prolongs atrial refractoriness by inhibition of a background K$^+$ channel

Marieke W. Veldkamp[1], Guillaume S.C. Geuzebroek[2], Antonius Baartscheer[1], Arie O. Verkerk[3], Cees A. Schumacher[1], Gedeon G. Suarez[4], Wouter R. Berger[1], Simona Casini[1], Shirley C.M. van Amersfoorth[1], Koen T. Scholman[3], Antoine H.G. Driessen[5], Charly N.W. Belterman[1], Antoni C.G. van Ginneken[3], Joris R. de Groot[1], Jacques M.T. de Bakker[1], Carol Ann Remme[1], Bas J. Boukens[3] & Ruben Coronel[1,6]

The cardiac autonomic nervous system (ANS) controls normal atrial electrical function. The cardiac ANS produces various neuropeptides, among which the neurokinins, whose actions on atrial electrophysiology are largely unknown. We here demonstrate that the neurokinin substance-P (Sub-P) activates a neurokinin-3 receptor (NK-3R) in rabbit, prolonging action potential (AP) duration through inhibition of a background potassium current. In contrast, ventricular AP duration was unaffected by NK-3R activation. NK-3R stimulation lengthened atrial repolarization in intact rabbit hearts and consequently suppressed arrhythmia duration and occurrence in a rabbit isolated heart model of atrial fibrillation (AF). In human atrial appendages, the phenomenon of NK-3R mediated lengthening of atrial repolarization was also observed. Our findings thus uncover a pathway to selectively modulate atrial AP duration by activation of a hitherto unidentified neurokinin-3 receptor in the membrane of atrial myocytes. NK-3R stimulation may therefore represent an anti-arrhythmic concept to suppress re-entry-based atrial tachyarrhythmias, including AF.

[1] Department of Clinical and Experimental Cardiology, Heart Center, Academic Medical Center, Meibergdreef 9, 1105 AZ Amsterdam, The Netherlands. [2] Department of Cardiothoracic Surgery, RadboudUMC, Geert Grooteplein Zuid 10, 6525 GA Nijmegen, The Netherlands. [3] Department of Medical Biology, Academic Medical Center Amsterdam, Meibergdreef 9, 1105 AZ, Amsterdam, The Netherlands. [4] Biomedical Sciences VU University Medical Center, De Boelelaan 1105, 1081 HV Amsterdam, The Netherlands. [5] Department of Cardiothoracic Surgery, Heart Center, Academic Medical Center, Meibergdreef 9, 1105 AZ Amsterdam, The Netherlands. [6] L'Institut de RYthmologie et de Modélisation Cardiaque (LIRYC), Fondation Université Bordeaux, Avenue du Haut Lévêque-33604 Pessac cedex, Bordeaux, France. These authors contributed equally: Marieke W. Veldkamp, Guillaume S. C. Geuzebroek, Antonius Baartscheer, Arie O. Verkerk. Correspondence and requests for materials should be addressed to M.W.V. (email: m.w.veldkamp@amc.uva.nl)

Normal atrial electrophysiology is critically controlled by the cardiac autonomic nervous system (ANS), which modulates heart rate, excitability, conduction and refractoriness[1,2]. The cardiac ANS plays a key role in the genesis of malignant atrial arrhythmias, particularly atrial fibrillation (AF)[3–7]. The major neurotransmitters that mediate the control of cardiac electrophysiological properties are the classical neurotransmitters acetylcholine (ACh) and noradrenalin (NA). However, the intracardiac ganglia located in fat pads on the heart produce a variety of neuropeptides including substance-P (Sub-P)[8–11], in addition to NA and ACh. While the actions of these neuropeptides on the cardiac neuron as effector cell are subject of ongoing investigation, their direct effects on atrial cardiomyocyte electrophysiology are largely unknown. This is relevant considering the potential role of neuropeptides in arrhythmogenesis as they may constitute targets for anti-arrhythmic therapy. Current therapeutic approaches in AF are aimed at prolonging action potential duration and consequently increasing refractoriness of atrial tissue, thereby reducing the susceptibility for re-entrant based arrhythmias[12,13]. However, clinical success rate is often diminished by the limited selectivity for atrial tissue, potentially leading to ventricular pro-arrhythmia secondary to action potential prolongation in this cardiac compartment[12,13].

Substance-P[14] belongs to the tachykinin peptide family. It exerts its biological effects by binding to one or more of three distinct types of neurokinin receptors (NK-1R, NK-2R and NK-3R), with a preferential binding for NK-1R[15,16]. Sub-P is widely distributed in the cardiovascular system and is involved in the regulation of many physiological functions, including heart rate (through stimulation of cholinergic neurons) and blood pressure[17,18]. Reduced Sub-P content has been associated with AF incidence[19,20]. Specifically, the occurrence of post-operational AF was associated with decreased Sub-P serum levels in patients who had undergone coronary artery bypass surgery[19]. Furthermore, degeneration of Sub-P immuno-reactive nerves has been reported in a dog model of AF[20]. These observations imply that intact Sub-P levels protect against the development of AF. However, a direct action of Sub-P on atrial myocardial electrophysiology has thus far not been identified.

In this study, we investigated the electrophysiological effects of the neuropeptide Sub-P in isolated rabbit atrial myocytes, and in Langendorff-perfused and in situ rabbit hearts. We show that Sub-P produces a selective prolongation of atrial AP duration and refractoriness in a dose-dependent manner. Importantly, these effects were also present at high pacing rates (typically observed in AF), and were absent in ventricular myocardium, indicating atrial-specific efficacy. We demonstrate that these effects are mediated through activation of the neurokinin-3 receptor (NK-3R) and involve inhibition of a background K$^+$ current. In a rabbit isolated heart model of AF, NK-3R stimulation produced a strong anti-arrhythmic effect by reducing AF duration and incidence. We therefore propose NK-3R as a therapeutic target for re-entry-based atrial arrhythmias, including AF.

## Results

### Sub-P prolongs the action potential in atrial cardiomyocytes.

Figure 1 illustrates that application of substance-P (Sub-P; 1 µM) resulted in significant action potential (AP) prolongation (Fig. 1a) in an isolated rabbit atrial cardiomyocyte. The average effects (mean ± SEM) of Sub-P (1 µM) on AP duration at 20%, 50% and 90% of repolarization (APD$_{20}$, APD$_{50}$ and APD$_{90}$) in cardiomyocytes stimulated at 1 Hz are summarized in Fig. 1b. APD$_{50}$ (Control: $21.2 \pm 7.5$ vs. Sub-P: $43.4 \pm 14.6$ ms, $p < 0.05$) and APD$_{90}$ (Control: $89.0 \pm 2.0$ vs. Sub-P: $125.2 \pm 2.5$ ms, $p < 0.001$)

were statistically significantly increased following Sub-P application. Sub-P furthermore reduced the resting membrane potential (RMP) (Control: $-81.0 \pm 1.5$ vs. Sub-P: $-75.0 \pm 2.1$ mV, $p < 0.05$, Fig. 1c, Table 1) and the action potential amplitude (APA) (Control: $108.6 \pm 3.4$ mV vs. Sub-P: $97.0 \pm 6.1$ mV, $p < 0.05$, Fig. 1c, Table 1). A trend towards a concomitant reduction in AP upstroke velocity ($V_{max}$) was observed following Sub-P application (Fig. 1d, Table 1), but this did not reach statistical significance due to a large variation between cells.

Sub-P prolonged APD$_{90}$ in a dose-dependent manner (Fig. 1e). The increase in APD$_{90}$ was statistically significant at a concentration of 10 nM Sub-P and higher, and exceeded 20% at a concentration of 1 µM (Fig. 1e, dotted lines). Sub-P caused a significant AP prolongation at all tested stimulation frequencies (1, 2, 3 and 4 Hz, Fig. 1f).

### Effects of Sub-P on atrial membrane currents.

We next investigated the effects of Sub-P on the major membrane currents controlling atrial AP duration in rabbit atrial cardiomyocytes, including steady-state and transient outward currents, L-type calcium currents, and calcium-dependent currents.

### Steady-state and transient outward currents.

Representative examples of current tracings recorded at $-120$ and $+40$ mV (see protocol, Fig. 2a) show a decrease in steady-state outward current in the presence of Sub-P (10 µM), which was confirmed by the average current–voltage (I–V) relationships in Fig. 2b. A statistically significant decrease in steady-state outward current was induced at all voltages positive to $-50$ mV. At $+50$ mV the outward current decreased by 20% from $6.9 \pm 0.5$ pA/pF (Control) to $5.5 \pm 0.6$ pA/pF (Sub-P, $p < 0.05$). In addition, the inward component of steady-state current at $-120$ mV was moderately decreased by Sub-P (Control: $-6.8 \pm 1.1$ vs Sub-P: $-6.0 \pm 1.3$ pA/pF, $p < 0.05$).

The resultant I–V curve of the Sub-P sensitive current (Fig. 2c, grey filled circles) is virtually linear and has a reversal potential (dotted line) close to the calculated Nernst potential for K$^+$ ($-85$ mV), indicating that it concerns a background K$^+$ current.

For accurate measurement of the transient outward current (I$_{to}$), blockers of the Na$^+$ current (I$_{Na}$), the L-type calcium current (I$_{Ca,L}$), and of the slow and rapid components of the delayed rectifier currents (I$_{Kr}$ and I$_{Ks}$) were included in the various solutions (see Methods section). Figure 2d shows representative I$_{to}$ currents upon depolarization to $+50$ mV, demonstrating no change in peak currents following application of Sub-P (10 µM). On average, Sub-P did not affect I$_{to}$ peak densities (Fig. 2e) or the time course of current decay at $+50$ mV (Table 2), but it caused a small, but significant shift in $V_{1/2}$ of inactivation by $-3.5$ mV (Table 2, $p < 0.05$), as well as a steepening of the slope of the voltage dependency of activation (Table 2, $p < 0.01$). Figure 2f depicts the average I–V relationships, obtained by current amplitude measurement at the end of the 500 ms voltage steps. Despite the presence of the various ion channel blockers, Sub-P causes a decrease in the steady-state inward current from $-6.9 \pm 1.6$ pA/pF to $-5.8 \pm 1.5$ pA/pF at $-120$ mV ($p < 0.05$), and a 25% reduction in steady-state outward current at $+50$ mV (Control: $6.1 \pm 0.85$ vs. Sub-P: $4.6 \pm 0.46$ pA/pF, $p < 0.05$). The Sub-P-sensitive current under these experimental conditions is plotted in Fig. 2c (grey filled triangles). It shows an I–V relation similar to the sub-P-sensitive current in the absence of the various ion channel blockers (Fig. 2c, grey filled circles). These data demonstrate that the Sub-P effects on background current are not mediated by changes in the delayed rectifier potassium currents I$_{Kr}$ and I$_{Ks}$.

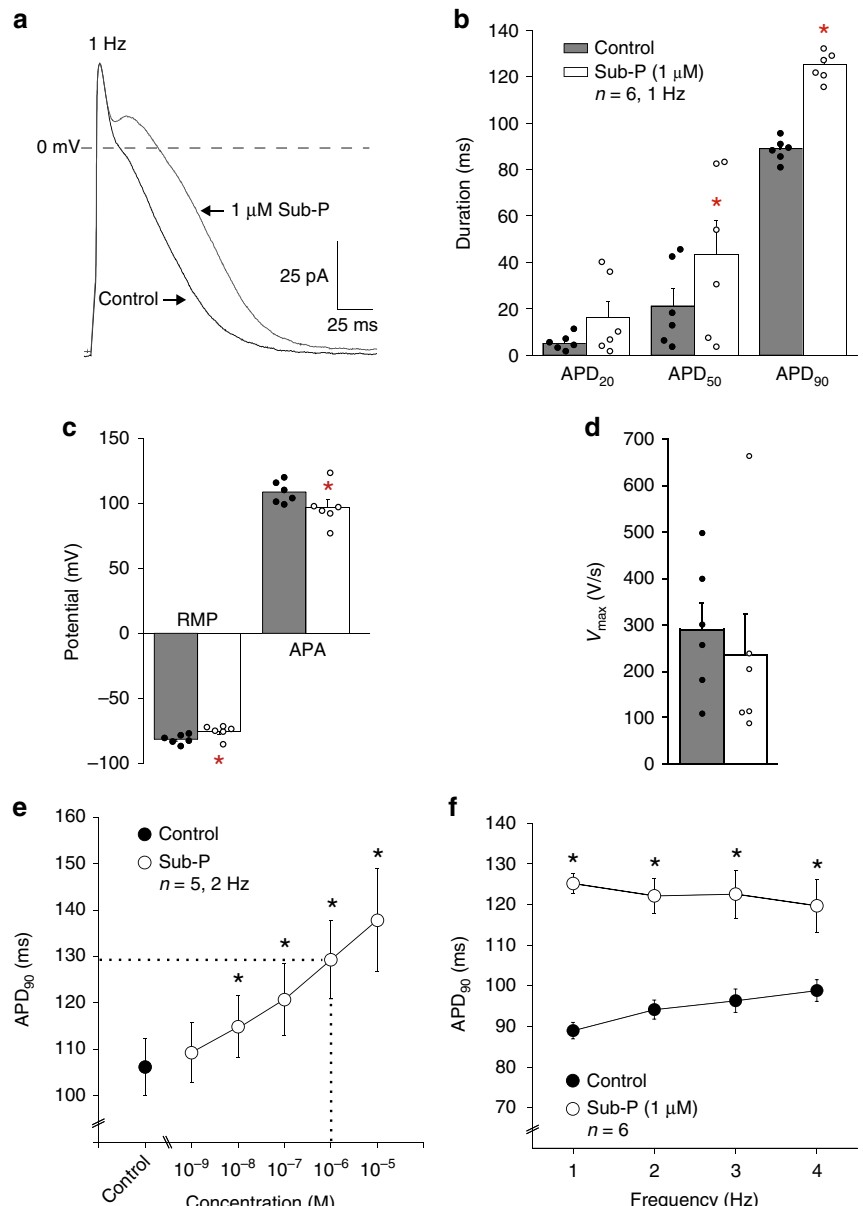

**Fig. 1** Effect of substance-P on action potential parameters of rabbit atrial cardiomyocytes. **a** APs elicited at 1 Hz from an atrial cardiomyocyte under control conditions and in the presence of 1 μM Sub-P. **b**–**d** Bar graph showing average values for AP duration at 20, 50 and 90% of repolarization (APD$_{20}$, APD$_{50}$ and APD$_{90}$) **b**, resting membrane potential (RMP) and action potential amplitude (APA) **c**, and maximal upstroke velocity ($V_{max}$) **d** before (Control) and after application of 1 μM Sub-P. ($N = 4$, $n = 6$, paired $t$-test.) **e** Concentration-dependence of APD$_{90}$ before (Control) and after application of increasing concentrations of Sub-P. Dotted lines: concentration of Sub-P at which the increase of APD$_{90}$ exceeded 20%. ($N = 4$, $n = 5$, one-way RM ANOVA.) **f** Frequency-dependence of APD$_{90}$ before (Control) and after application of 1 μM Sub-P ($N = 4$, $n = 6$, two-way RM ANOVA). All values shown are mean ± SEM. * $P < 0.05$

**L-type calcium current and calcium-dependent currents**. The representative $I_{Ca,L}$ recordings upon a depolarizing pulse to 0 mV (Fig. 3a, top: voltage protocol) show that peak $I_{Ca,L}$ is reduced following application of Sub-P (10 μM). The average I–V relationships of $I_{Ca,L}$ (Fig. 3b) confirm a statistically significant decrease in peak $I_{Ca,L}$ by about 20% from $-21 \pm 3.3$ pA/pF in Control to $-16.6 \pm 2.3$ pA/pF ($p < 0.05$) in the presence of Sub-P at 0 mV, expected to decrease AP duration (contrary to the observed AP prolonging effect of Sub-P). Neither the voltage dependencies of (in)activation, nor the time course of current decay at 0 mV (Table 2) were changed.

Supplementary Figure 1a shows representative examples of intracellular Ca$^{2+}_i$ transients in atrial cardiomyocytes following field stimulation at 5 Hz, before and after application of Sub-P (10 μM). On average, no differences in diastolic Ca$^{2+}_i$, systolic Ca$^{2+}_i$ or transient amplitude occurred (Supplementary Figure 1b).

Figure 3c shows a representative example of the Na$^+$–Ca$^{2+}$ exchange current ($I_{NCX}$) recorded during a descending voltage ramp protocol (top) before and after application of Sub-P (10 μM). On average, Sub-P had no direct effect on the forward (inward) or inverse (outward) mode of the $I_{NCX}$ (Fig. 3d).

The Ca$^{2+}$-activated Cl$^-$ current ($I_{Cl(Ca)}$) was defined as the early transient peak amplitude upon depolarizing steps to +30, +40 and +50 mV (Fig. 3e). Average $I_{Cl(Ca)}$ peak amplitudes were similar before (Control) and after application of Sub-P (Fig. 3f).

**Table 1 Effects of substance-P and selective NK-1, NK-2 and NK-3 receptor agonists on action potential parameters in atrial and ventricular myocytes**

|  | RMP (mV) | APA (mV) | $V_{max}$ (V/s) | $APD_{20}$ (ms) | $APD_{50}$ (ms) | $APD_{90}$ (ms) |
|---|---|---|---|---|---|---|
| *Atrial myocyte* |  |  |  |  |  |  |
| Control | −81.0 ± 1.5 | 108.6 ± 3.5 | 290.5 ± 57.9 | 5.1 ± 1.4 | 21.2 ± 7.5 | 89.0 ± 2.0 |
| Sub-P (1 µM, $n = 6$) | −75.0 ± 2.1[a] | 97.0 ± 6.1[a] | 236.7 ± 88.7 | 16.0 ± 7.0 | 43.4 ± 14.6[a] | 125.2 ± 2.5[a] |
| Control | −82.4 ± 1.1 | 106.0 ± 2.4 | 238.6 ± 35.2 | 4.8 ± 0.6 | 14.2 ± 3.1 | 73.7 ± 3.1 |
| [Sar9,Met(O2)11]-SP (100 nM, $n = 7$) | −79.9 ± 1.7 | 101.8 ± 5.4 | 252.1 ± 72.9 | 5.0 ± 0.3 | 18.8 ± 3.9 | 72.7 ± 4.3 |
| Control | −81.6 ± 1.1 | 111.5 ± 2.3 | 281.6 ± 57.6 | 4.1 ± 1.2 | 25.0 ± 13.3 | 98.4 ± 10.3 |
| (b-Ala)-NKA (4-10) (100 nM, $n = 4$) | −79.0 ± 2.2 | 100.9 ± 6.1 | 255.1 ± 87.0 | 6.2 ± 1.4 | 35.1 ± 14.7 | 105.2 ± 10.4 |
| Control | −82.7 ± 1.2 | 109.9 ± 2.6 | 259.9 ± 47.8 | 6.7 ± 1.3 | 27.9 ± 7.2 | 95.8 ± 11.3 |
| Senktide (100 nM, $n = 16$) | −80.6 ± 1.5 | 104.0 ± 4.3 | 259.7 ± 39.2 | 11.3 ± 2.9 | 64.5 ± 7.5[a] | 143.9 ± 9.0[a] |
| *Ventricular myocyte* |  |  |  |  |  |  |
| Control | −85.9 ± 1.2 | 121.6 ± 1.6 | 260.4 ± 55.3 | 67.0 ± 9.4 | 140.4 ± 11.6 | 176.0 ± 10.6 |
| Sub-P (1 µM, $n = 9$) | −86.5 ± 1.2[a] | 122.3 ± 1.8 | 300.0 ± 53.0[a] | 65.9 ± 9.4 | 137.5 ± 11.3 | 171.8 ± 10.1 |
| Control | −86.4 ± 0.4 | 120.8 ± 1.3 | 267.7 ± 45.9 | 59.8 ± 7.9 | 134.3 ± 8.3 | 170.4 ± 7.4 |
| Senktide (100 nM, $n = 10$) | −86.7 ± 0.6 | 121.1 ± 1.4 | 298.0 ± 41.6[a] | 56.8 ± 9.4 | 128.9 ± 10.4 | 166.2 ± 8.6 |

All values are mean ± SEM

RMP: resting membrane potential, APA: action potential amplitude, $V_{max}$: upstroke velocity, $APD_{20}$: action potential duration at 20% repolarization, $APD_{50}$: action potential duration at 50% repolarization, $APD_{90}$: action potential duration at 90% repolarization

Paired $t$-test = $p < 0.05$; Compared to control = [a]

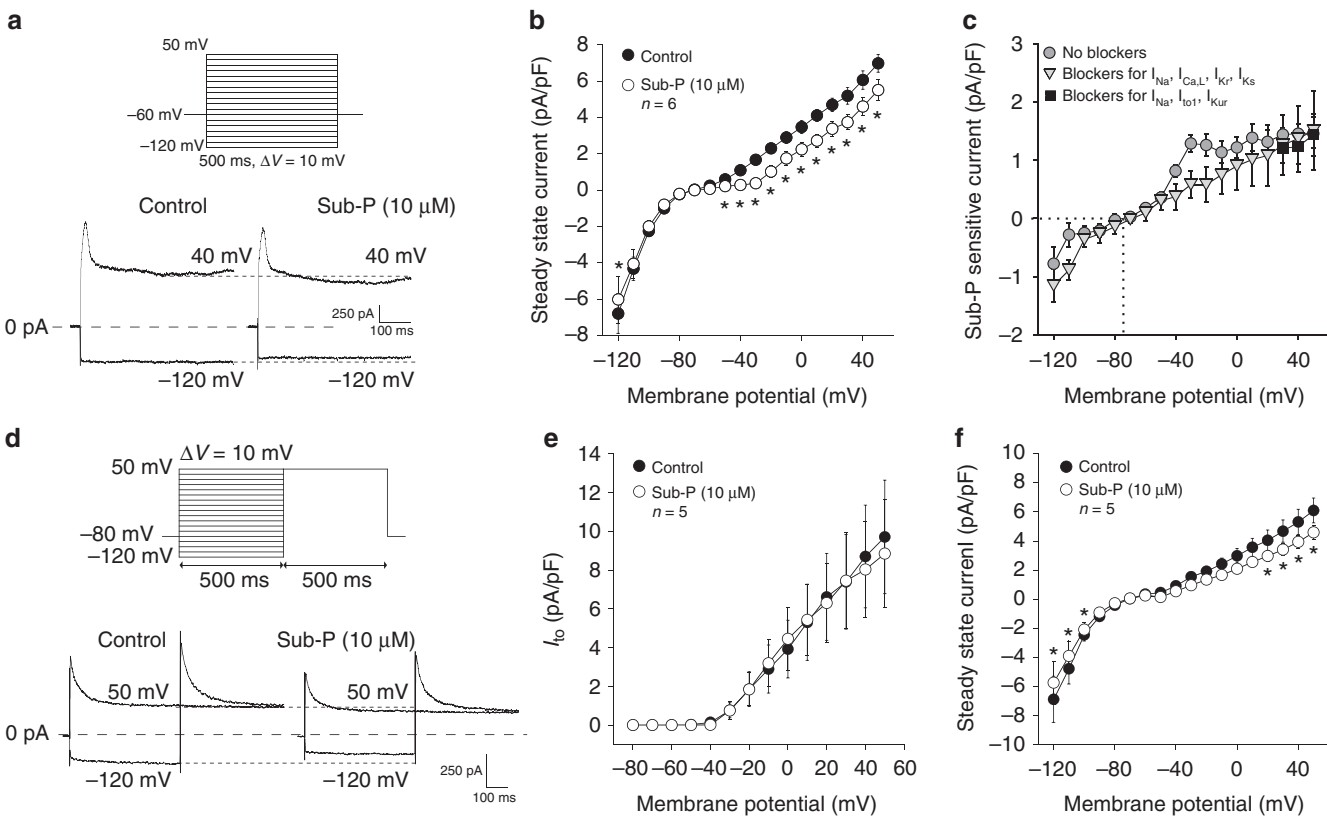

**Fig. 2** Effect of substance-P on steady-state and transient outward currents. **a** Voltage protocol and current tracings recorded at −120 and +40 mV before (Control) and after application of 10 µM Sub-P. **b** Average current–voltage relationships of the steady-state current in Control and after application of Sub-P (10 µM) ($N = 2$, $n = 6$, two-way RM ANOVA). **c** Average current–voltage relationships of the Sub-P sensitive current. Sub-P sensitive currents were derived from **b** (grey filled circles), **f** (grey filled triangles) and Supplementary Figure 2 (black filled squares), as differential current. **d** Voltage protocol and transient outward K$^+$ current (I$_{to}$) current tracings recorded at −120 and +50 mV before (Control) and after application of Sub-P (10 µM). **e** Average current–voltage (I–V) relationships of I$_{to}$ peak currents before (Control) and after application of Sub-P (10 µM) ($N = 4$, $n = 5$, two-way RM ANOVA). **f** Average I–V relationships of steady-state current before (Control) and after application of Sub-P (10 µM) ($N = 4$, $n = 5$, two-way RM ANOVA). Blockers for Na$^+$, L-type Ca$^{2+}$, rapid and slow delayed rectifier K$^+$ currents were present. All values shown are mean ± SEM. *$P < 0.05$

**Table 2 Effects of substance-P (10 µM) on biophysical properties of the L-type calcium current ($I_{Ca,L}$) and the transient outward current ($I_{to}$)**

| | $I_{Ca,L}$-Control | $I_{Ca,L}$-Sub-P | $n$ | $I_{to}$-Control | $I_{to}$-Sub-P | $n$ |
|---|---|---|---|---|---|---|
| Current density (pA/pF) | $-21.3 \pm 3.3$ | $-16.6 \pm 2.3^a$ | 6 | $9.7 \pm 2.9$ | $8.9 \pm 2.8$ | 5 |
| *Activation* | | | | | | |
| $V_{1/2}$, mV | $-15.4 \pm 0.8$ | $-17.2 \pm 1.4$ | 6 | $-7.0 \pm 7.4$ | $-15.1 \pm 3.3$ | 5 |
| $k$, mV | $6.8 \pm 0.5$ | $7.0 \pm 0.7$ | 6 | $14.2 \pm 0.9$ | $9.8 \pm 0.1^a$ | 5 |
| *Inactivation* | | | | | | |
| $V_{1/2}$, mV | $-28.9 \pm 1.3$ | $-31.2 \pm 2.3$ | 6 | $-32.5 \pm 3.4$ | $-35.9 \pm 2.8^a$ | 5 |
| $k$, mV | $4.4 \pm 0.3$ | $4.9 \pm 0.3$ | 6 | $-5.9 \pm 0.9$ | $-5.4 \pm 0.2$ | 5 |
| *Inactivation time course* | | | | | | |
| $\tau_{fast}$, ms | $4.4 \pm 0.6$ | $4.9 \pm 0.7$ | 6 | $39.6 \pm 13.2$ | $41.3 \pm 8.1$ | 5 |
| $\tau_{slow}$, ms | $28.0 \pm 3.5$ | $28.2 \pm 3.4$ | 6 | $131.1 \pm 33.7$ | $114.6 \pm 16.2$ | 5 |

All values are mean ± SEM. Two-way repeated measures ANOVA followed by pairwise comparison using the Student–Newman–Keuls method = $p < 0.05$; Compared to control = $^a$
Sub-P: substance-P, $V_{1/2}$: voltage of half-maximal (in)activation, $k$: slope factor, $t_{fast}$ and $t_{slow}$: fast and slow time constants of inactivation

However, Sub-P reduced steady-state outward currents at the end of the 250 ms depolarizing steps by ~25% (Supplementary Figure 2; Control: $7.2 \pm 0.8$ pA/pF vs. Sub-P: $5.7 \pm 0.5$, $V_m = +50$ mV, $p < 0.05$). Data points for the Sub-P-sensitive current at +30, +40 and +50 mV are included in the I–V relation of Fig. 2c (black filled squares), and show complete overlap with the Sub-P-sensitive current in the absence of ion channel blockers. Since $I_{Cl(Ca)}$ measurements were performed in the presence of 2 mM 4-AP, the effect of Sub-P is not mediated by a reduction in sustained component of $I_{to}$ or $I_{Kur}$.

Taken together, these data indicate that Sub-P acts through inhibition of a $K^+$ background current in rabbit atrial myocytes.

**Neurokinin-3 receptor mediates action potential prolongation.** To define the neurokinin receptor (NK-R) subtype mediating the AP prolonging effect produced by Sub-P, we examined the effect of various NK receptor subtype selective agonists on AP duration in isolated rabbit atrial cardiomyocytes. Fig. 4a, b (left panels) show representative APs at 2 Hz before (Control) and after application of the NK-1R agonist [Sar9,Met(O2)11]-SP (100 nM), and the NK-2R agonist (β-Ala)-NKA(4-10) (100 nM), respectively. Administration of NK-1R and NK-2R agonists did not significantly alter AP morphology or duration (Fig. 4a, b (right panels), Table 1). In contrast, application of 100 nM of the NK-3R agonist Senktide (Fig. 4c) significantly prolonged $APD_{50}$ (Control: $27.9 \pm 7.2$ vs. Senktide: $64.5 \pm 7.5$ ms, $p < 0.001$) and $APD_{90}$ (Control: $95.8 \pm 11.3$ vs. Senktide: $143.9 \pm 9.0$ ms, $p < 0.001$). Senktide (100 nM) prolonged $APD_{50}$ and $APD_{90}$ by 130% and 50%, respectively, but did not alter other AP parameters (Table 1). To rule out the possibility of a time-dependent increase in $APD_{90}$ contributing to the observed Senktide effect, we performed a set of time-matched controls. Supplementary Figure 3a shows in a representative example that the $APD_{90}$ remained stable during a period of 15 min, whereafter the application of 50 nM Senktide caused a steep increase in $APD_{90}$. On average, $APD_{90}$ remained unchanged at 1, 5 and 10 min (common duration of the experiment) (Supplementary Figure 3b).

Figure 4d shows the dose-response relationship of Senktide and atrial $APD_{90}$. At a concentration of ~10 nM, Senktide induced a 20% increase in $APD_{90}$ (dotted lines), indicating that it is 100 times more potent than Sub-P (cf. Fig. 1e). Taken together these data indicate that AP prolongation by Sub-P is consequent to a reduction in a background $K^+$ current which is mediated through stimulation of the NK-3 receptor. This was further supported by the observation that the Senktide-induced AP prolongation was largely prevented by pre-incubation with the competitive NK-3R

antagonist Osanetant (300 nM) (Fig. 4d). As the AP prolonging effect of Sub-P is entirely mediated through the NK-3R, we maintained the use of Senktide—a potent analogue of the endogenous agonist of the NK-3R (neurokinin B)—in all following experiments.

**Nature of the background $K^+$ current.** We next explored the nature of the background $K^+$ current affected by NK-3 receptor stimulation. Two-pore-domain potassium (K2P) channels have been often shown to underlie the background $K^+$ currents in excitable tissues[21]. The NK-3 receptor is coupled to the Gq/11 subgroup of G-proteins[22], and several members of the K2P family are known to be potently inhibited by receptors that signal through Gq/11[23–25]. Of those, TREK-1, TASK-1 and TASK-3, have been shown to be functionally relevant for atrial AP duration[26–28]. We therefore considered these 3 channels likely candidates for the AP prolonging effect of NK-3 receptor stimulation. We investigated mRNA expression levels of the different members of the K2P channel family as well as the various NK receptor types in rabbit left atrial myocardium using RNA sequencing. As shown in Fig. 5a, mRNA expression levels of *TACR3* (encoding NK-3R) were clearly higher than those of *TACR1* and *TACR2* (encoding NK-1R and NK-2R, respectively). We did not find differences in mRNA expression levels for the different genes between the left atrial free wall and the left atrial appendage (Fig. 5a, upper right inset). The presence of NK-3R on the plasma membrane of rabbit atrial cardiomyocytes was confirmed by immunohistochemistry (Fig. 5b). The RNA sequencing results furthermore demonstrated higher expression levels of *KCNK3* (encoding TASK-1) in rabbit atrial tissue as compared to *KCNK2* (encoding TREK-1) and *KCNK9* (encoding TASK-3) (Fig. 5a).

To test the potential contribution of these K2P channels to the observed effects of NK-3 receptor stimulation, we assessed the effects of selective blockers for these channels on steady-state current and AP duration. We first examined the effects of $ZnCl_2$ on steady-state outward currents in individual rabbit atrial cardiomyocytes. Zinc has been reported to be a selective blocker of TASK-3 channels, but may also inhibit TASK-1 and TREK-1 channels, albeit with lower sensitivities.[29–31] As expected, the average I–V relation in Fig. 6b shows a significant decrease in steady-state outward current at all voltages positive to $-60$ mV following Senktide application (50 nM), similar to the effects observed for Sub-P (Fig. 2b). At +30 mV the outward current decreased by 44% from $4.6 \pm 0.4$ pA/pF (Control, $n = 8$) to $3.2 \pm 0.3$ pA/pF (Senktide, $n = 8$, $p < 0.05$). The Senktide-sensitive I–V relation (Fig. 6c) is similar to the Sub-P-sensitive current (Fig. 2c),

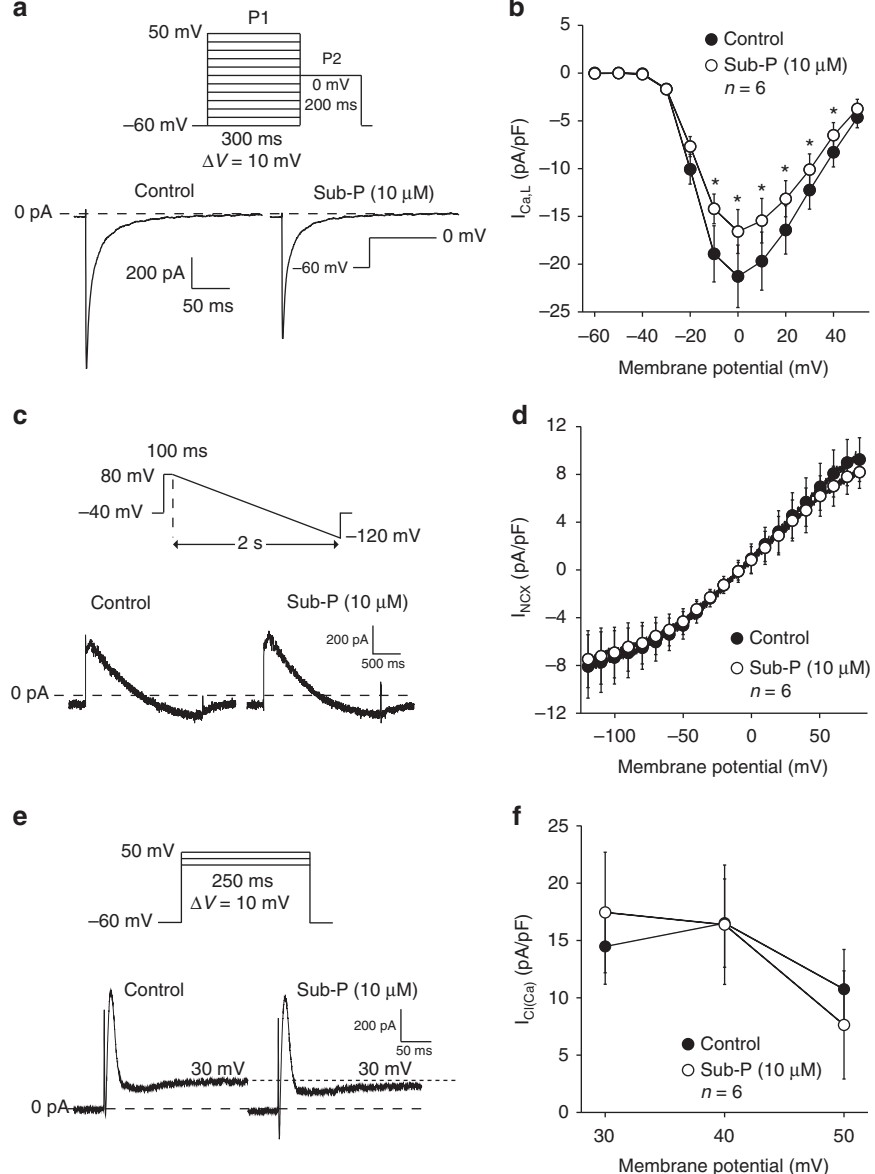

**Fig. 3** Effect of substance-P on L-type $Ca^{2+}$ current and calcium-dependent currents. **a** Voltage protocol and $I_{Ca,L}$ tracings recorded at 0 mV under control conditions (Control) and in the presence of Sub-P (10 μM). **b** Average current–voltage (I–V) relationships of $I_{Ca,L}$ during Control and after application of Sub-P (10 μM) ($N = 3$, $n = 6$, two-way RM ANOVA). **c** Voltage protocol and $Na^+$–$Ca^{2+}$ exchange current ($I_{NCX}$) tracings recorded in Control and in the presence of Sub-P (10 μM). **d** Average I–V relationships of $I_{NCX}$ before (Control) and after application of Sub-P (10 μM) ($N = 2$, $n = 6$, two-way RM ANOVA). **e** Voltage protocol and $Ca^{2+}$ activated $Cl^-$ current ($I_{Ca(Cl)}$) recordings at +30 mV in Control and in the presence of Sub-P (10 μM). **f** Average I–V relationships of peak $I_{Ca(Cl)}$ before (Control) and after application of Sub-P (10 μM) ($N = 2$, $n = 6$, two-way RM ANOVA). Blockers for $Na^+$-, transient outward and ultra-rapid delayed rectifier $K^+$ currents were continuously present. All values shown are mean ± SEM. *$P < 0.05$

both in shape and magnitude of current density. The application of $ZnCl_2$ significantly reduced steady-state outward current (Fig. 6d,e). At +30 mV, the average steady-state outward current (Fig. 6e) was reduced by 39% from 7.8 ± 0.9 pA (Control, $n = 5$) to 5.6 ± 0.8 ($ZnCl_2$, $n = 5$, $p < 0.05$). The $ZnCl_2$-sensitive current (Fig. 6f) was not critically different from the Senktide-sensitive current (Fig. 6c). In the continued presence of $ZnCl_2$, additional application of 50 nM Senktide did not further reduce steady-state outward current (Fig. 6f). These findings imply that the $ZnCl_2$-sensitive current and the Senktide-sensitive current are the same, and possibly carried by either TASK-1, TASK-3 or TREK-1 channels. The selective blockers for TASK-1 (ML-365), TASK-3 (PK-THPP) and TREK-1 (Spadin), all gave a minor prolongation of the $APD_{90}$ (Fig. 6g–i). Application of ML-365 gave rise to a 5%

increase in $APD_{90}$ from 95.6 ± 10.5 ms to 100.4 ± 10.9 ms (Fig. 6j, $n = 8$, $p < 0.05$), PK-THPP application induced an 8% increase in $APD_{90}$ from 73.5 ± 3.2 ms to 79.0 ± 3.1 ms (Fig. 6k, $n = 7$, $p < 0.05$), and Spadin application resulted in a 3% increase in $APD_{90}$ from 80.1 ± 6.7 ms to 82.1 ± 6.6 ms (Fig. 6l, $n = 7$, $p = $ NS) (Supplementary Table 1). Subsequent addition of 50 nM Senktide in the continued presence of the respective blockers, resulted in a 41% (ML-365), 53% (PK-THHP) and 42% (Spadin) increase in $APD_{90}$ (Fig. 6g–l, Supplementary Table 1). These percentages increase in $APD_{90}$ induced by Senktide in the presence of the selective blockers, is in the same order of magnitude as the increase in $APD_{90}$ by Senktide alone (Fig. 4d). This implies that the background current inhibited by NK-3 receptor stimulation is likely not carried by one of these $K^+$ channels alone.

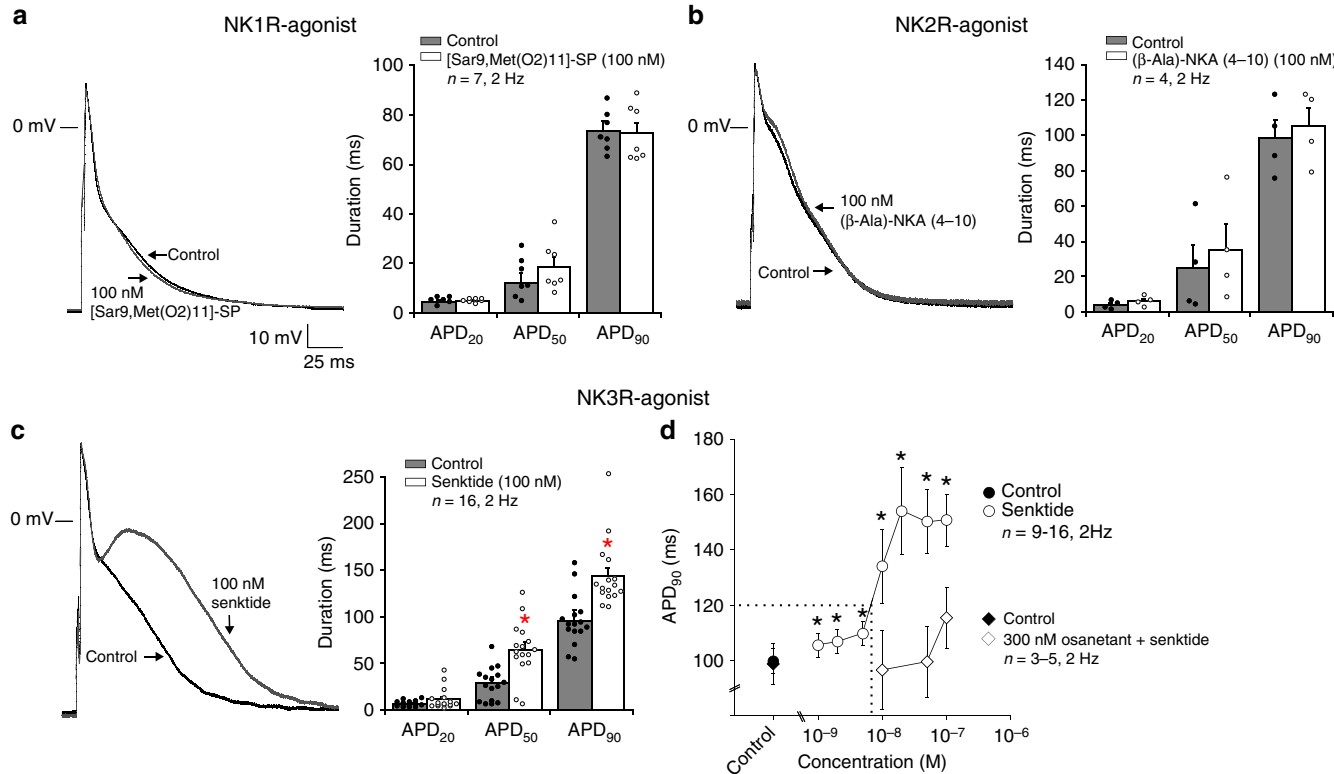

**Fig. 4** Effects of different selective neurokinin receptor agonists on action potential duration. **a** APs elicited at 2 Hz from an atrial cardiomyocyte under control conditions (Control) and in the presence of 100 nM NK-1R agonist [Sar9,Met(O2)11]-SP. Bar graph shows average values for AP duration at 20, 50 and 90% of repolarization ($APD_{20}$, $APD_{50}$ and $APD_{90}$) before (Control) and after application of 100 nM [Sar9,Met(O2)11]-SP ($N = 6$, $n = 7$, paired $t$-test). **b** Atrial APs elicited at 2 Hz in Control and in the presence of the NK-2R agonist (β-Ala)-NKA(4-10) (100 nM). Bar graph shows average values for $APD_{20}$, $APD_{50}$, and $APD_{90}$ before (Control) and after application of 100 nM (β-Ala)-NKA(4-10) ($N = 3$, $n = 4$, paired $t$-test). **c** Atrial APs elicited at 2 Hz in Control and in the presence of the NK-3R agonist Senktide (100 nM). Bar graph shows average values for $APD_{20}$, $APD_{50}$ and $APD_{90}$ before (Control) and after application of 100 nM Senktide ($N = 7$, $n = 16$, paired $t$-test). **d** Concentration-response curve for increase of APD induced by Senktide in the absence ($N = 13$, $n = 9-16$, one-way RM ANOVA) and presence of 300 nM Osanetant ($N = 3$, $n = 3-5$, two-way RM ANOVA). Dotted lines: concentration of Senktide at which $APD_{90}$ prolongation exceeded 20%. All values shown are mean ± SEM. *$P < 0.05$

**Ventricular APD is unaffected by NK-3 receptor stimulation.** We also explored the potential atrial selectivity of NK-3R stimulation, a prerequisite for safe anti-arrhythmic efficacy. We therefore tested the effects of Sub-P and Senktide on isolated rabbit ventricular cardiomyocytes. Figure 7a, b shows representative ventricular APs at 2 Hz in the absence and in the presence of the NK-3R agonists Sub-P (1 μM) and Senktide (100 nM), respectively. Neither Sub-P nor Senktide prolonged the ventricular AP. Overall, no relevant alterations were observed for the other ventricular AP characteristics on NK-3R stimulation, except for an increase in $V_{max}$ (Table 1, Fig. 7a, b).

**NK-3 receptor stimulation prevents atrial fibrillation.** To assess the potential impact of selective NK-3R activation on the in vivo, whole heart and tissue level, we tested whether NK-3R mediated AP prolongation is accompanied by an increase in effective refractory period (ERP) in rabbit intact hearts and human atrial tissue.

Figure 8a (left panel) shows atrial unipolar electrograms recorded from a Langendorff-perfused rabbit heart in response to programmed stimulation at the right atrium, showing a prolongation of right atrial effective refractory period (AERP) after addition of the NK-3R agonist (Senktide 20 nM). On average, right AERP increased from 68.1 ± 5.1 ms at baseline (Control) to 89.3 ± 6.3 ms after application of 20 nM Senktide (32% increase; $p = 0.00006$) (Fig. 8a, right panel). Left AERP, measured during programmed stimulation at the left atrium,

showed a similar increase in AERP in response to 20 nM Senktide (Supplementary Figure 5a, b).

We next studied the in vivo response to intravenous Senktide administration on AERP in open-chested rabbits. Supplementary Figure 5c shows unipolar electrograms recorded from the left atrium during programmed stimulation, showing an increased AERP following infusion of a single bolus of Senktide (11 nmol/kg). On average, AERP increased from 96.7 ± 13.3 ms to 130.0 ± 20.8 ms after application of Senktide (34% increase; Supplementary Fig. 5d, $p < 0.01$).

Similarly, we tested the effect of Senktide on ERP in human isolated left atrial appendages (LAA) resected from patients with persistent AF undergoing thoracoscopic surgery for AF. Similar to rabbit atrial tissue, robust cardiomyocyte plasma membrane labelling of NK-3R was observed in human LAA tissue (Fig. 5b). As shown in the example depicted in Fig. 8b (left panel), superfusion of the LAA preparation (stimulated at BCL 600 ms) with 100 nM Senktide led to an increase in ERP. Prolongation of the ERP occurred in all five human LAAs tested. On average, Senktide caused an 18.0 ± 0.1% increase of LAA ERP from 201.8 ± 6.1 ms (Control) to 237.8 ± 7.5 ms (Senktide) (Fig. 8b, right panel, $n = 5$, $p = 0.06$).

In a rabbit isolated heart model of AF based on atrial dilatation, we finally examined whether the increase in AERP resulting from NK-3 receptor stimulation by Senktide, suppresses AF inducibility. Fig. 8c shows representative electrograms recorded from the left atrium following a burst pacing protocol to provoke AF.

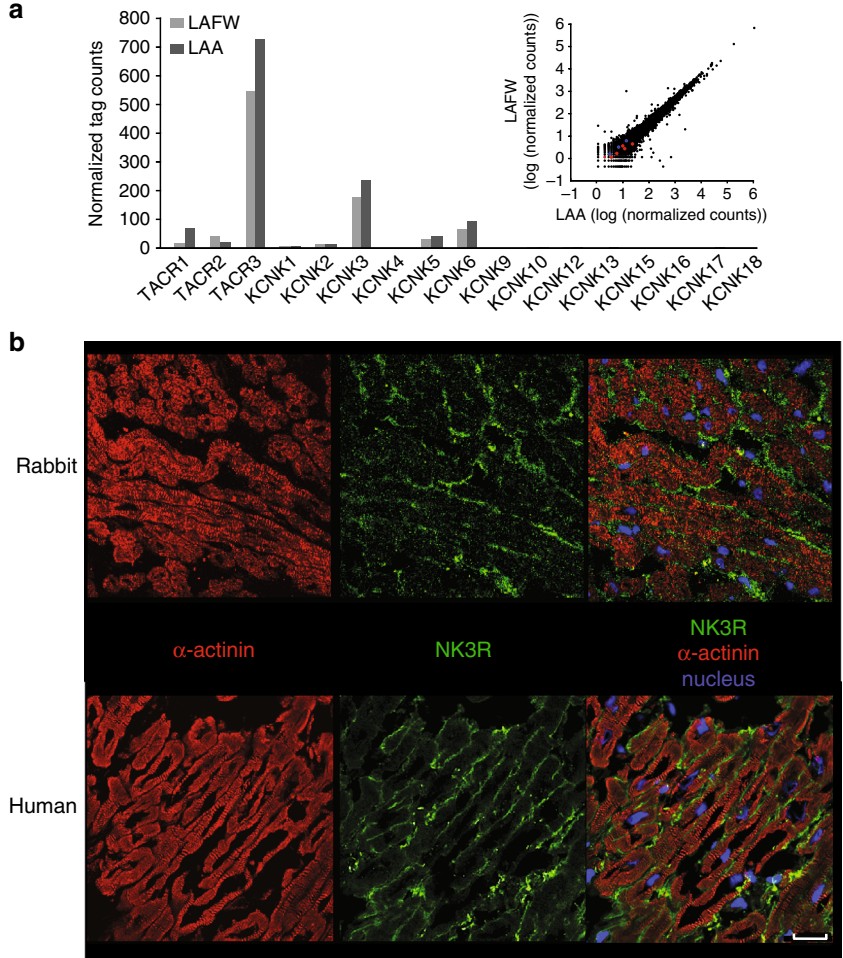

**Fig. 5** Expression of the NK-3 receptor and *KCNK* gene family in left atrium of rabbit and human. **a** Bar graph showing the normalized tag counts from RNA-seq for the *TACR* gene and *KCNK* gene family in the left atrial free wall (LAFW) and the left atrial appendage (LAA) of rabbit ($n = 1$). Right inset: Scatter plot showing the relation between expression profiles of all genes in LAWF and LAA in rabbit. The blue and red dots indicate *TACR* and *KCNK* genes, respectively. **b** Sections of rabbit (upper panel) and human (lower panel) left atrium showing labelling of the NK-3R (green) in the peripheral sarcolemma of cardiomyocytes, co-stained for the intracellular myocardial marker a-actinin (red). Scale bar 25 μM. Negative controls for immunohistochemistry are shown in Supplementary Figure 4

All electrograms were recorded from the same heart with and without increased left atrial pressure, and in the absence and presence of Senktide. In the non-dilated left atrium (0 cm $H_2O$), burst pacing incidentally elicited short runs of premature atrial activations (Fig. 8c, upper left). However, in the dilated atrium where intra-atrial pressure was increased to 20 cm $H_2O$, burst pacing regularly resulted in episodes of AF (Fig. 8c, lower left). On average, the total duration of arrhythmia episodes following each of the 10 consecutive runs of burst pacing, increased from $4.0 \pm 0.6$ s at 0 cm $H_2O$ to $20.6 \pm 5.3$ s at 20 cm $H_2O$ (Fig. 8d, $n = 7$, $p < 0.05$, Supplementary Table 2). In the presence of 20 nM Senktide, however, burst pacing failed to evoke any arrhythmias in the undilated atrium (Fig. 8c, upper right), and the duration of the arrhythmia episode was strongly reduced in the dilated atrium (Fig. 8c, lower right). On average, the total duration of arrhythmia episodes in the presence of 20 nM Senktide decreased by ~65% from $4.0 \pm 0.6$ s to $1.5 \pm 0.2$ s at 0 cm $H_2O$, and from $20.6 \pm 5.3$ s to $6.6 \pm 2.5$ s at 20 cm $H_2O$ (Fig. 8d, $n = 7$, $p < 0.05$, Supplementary Table 2). Senktide not only reduced the duration of the arrhythmia episodes, but also reduced the AF incidence. Figure 8e shows that AF incidence in the presence of Senktide was reduced from $0.08 \pm 0.04$ to $0.0 \pm 0.0$ ($n = 7$, $p < 0.05$) at 0 cm $H_2O$ and from $0.59 \pm 0.09$ to $0.17 \pm 0.08$ at 20 cm $H_2O$ ($n = 7$, $p < 0.05$),

representing a 70% reduction in AF vulnerability. All effects were reversible upon wash-out of Senktide, and on normalization of pressure (Fig. 8d, e, Supplementary Table 2). Thus, by selectively increasing AERP, NK-3R stimulation strongly reduces AF inducibility and duration.

## Discussion

This study provides biophysical and pharmacological evidence that the neuropeptide Sub-P prolongs the atrial action potential in rabbit myocytes by activation of the neurokinin-3 receptor (NK-3R) and consequent inhibition of a background $K^+$ current. We further demonstrate that the NK-3R mediated AP prolongation is maintained at high pacing rates and results in increased atrial effective refractory period (ERP) in Langendorff-perfused and in situ rabbit hearts. Also in human atrial appendages the latter phenomenon was observed. Ventricular action potential does not change following NK-3R stimulation, indicating atrial-specific efficacy. In a rabbit isolated heart model of AF, NK-3R stimulation exerts a potent anti-arrhythmic action by strongly reducing AF duration and incidence.

The AP prolonging effect by Sub-P was dose-dependent and occurred at nanomolar concentrations, supporting the view that

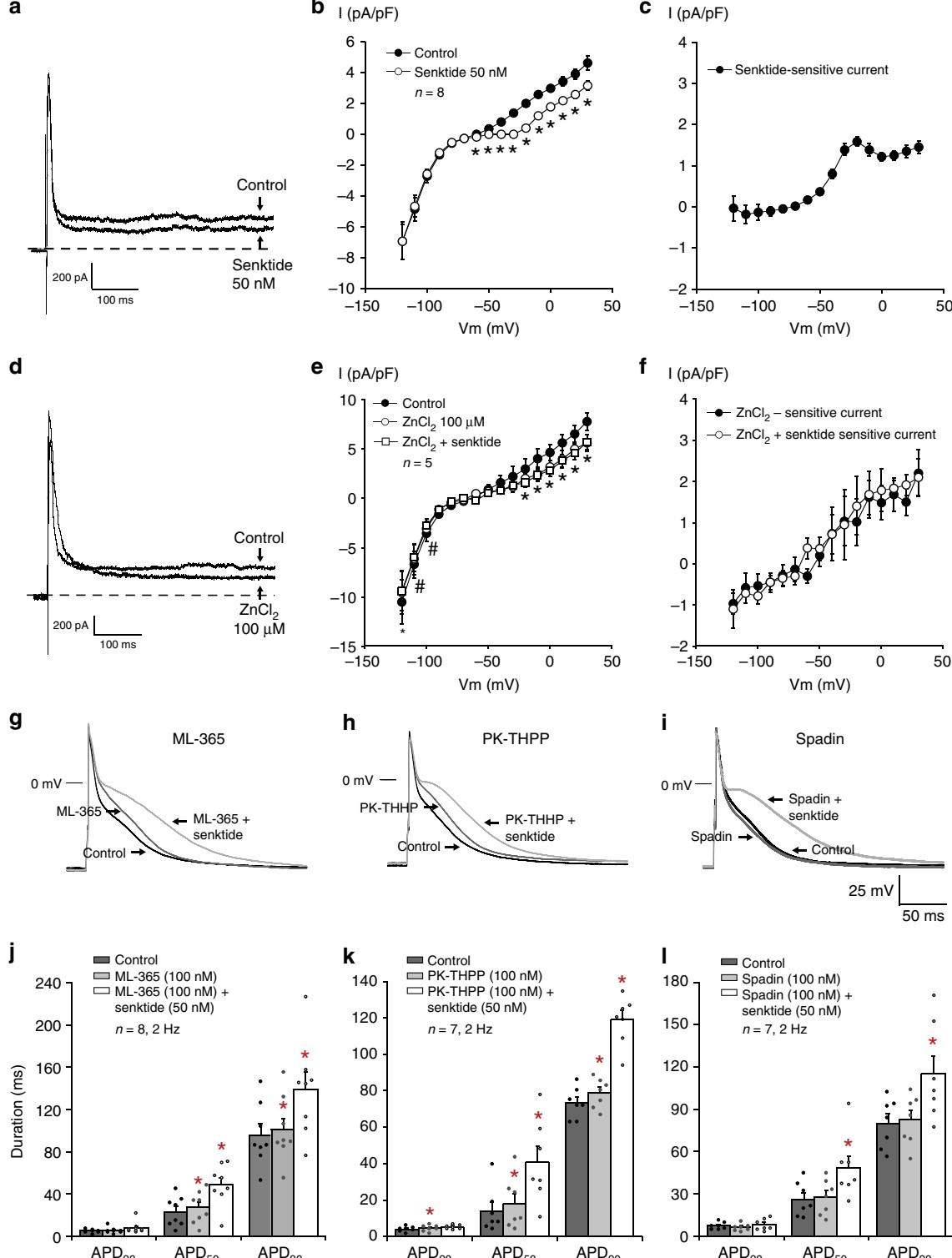

**Fig. 6** Effects of Senktide and different K2P channel blockers on atrial electrophysiology. **a–c** Current tracings recorded at +40 mV before (Control) and after application of 50 nM Senktide (**a**), associated average current–voltage I–V relationships of the steady-state current (**b**), and I–V relationships of Senktide-sensitive current (**c**) ($N = 4$, $n = 8$, two-way RM ANOVA). **d–f** Current tracings recorded at +40 mV before (Control) and after application of $ZnCl_2$ (100 µM), followed by additional Senktide (50 nM) (**d**), associated average I–V relationships of the steady-state current (**e**), and average I–V relationships of $ZnCl_2$-sensitive current (**f**). ($N = 4$, $n = 5$, two-way RM ANOVA). **g–i** Atrial action potentials (AP) elicited at 2 Hz under control conditions (Control) and in the presence of 100 nM ML-365 (**g**), PK-THPP (**h**) and Spadin (**i**). **j–l** Bar graphs showing average values for AP duration at 20, 50 and 90% of repolarization ($APD_{20}$, $APD_{50}$ and $APD_{90}$) before (Control) and after application of 100 nM ML-365 (**j**, $N = 4$, $n = 8$), PK-THPP (**k**, $N = 3$, $n = 7$), and Spadin (**l**, $N = 2$, $n = 7$). One-way RM ANOVA. All values shown are mean ± SEM. *$P < 0.05$ vs. Control

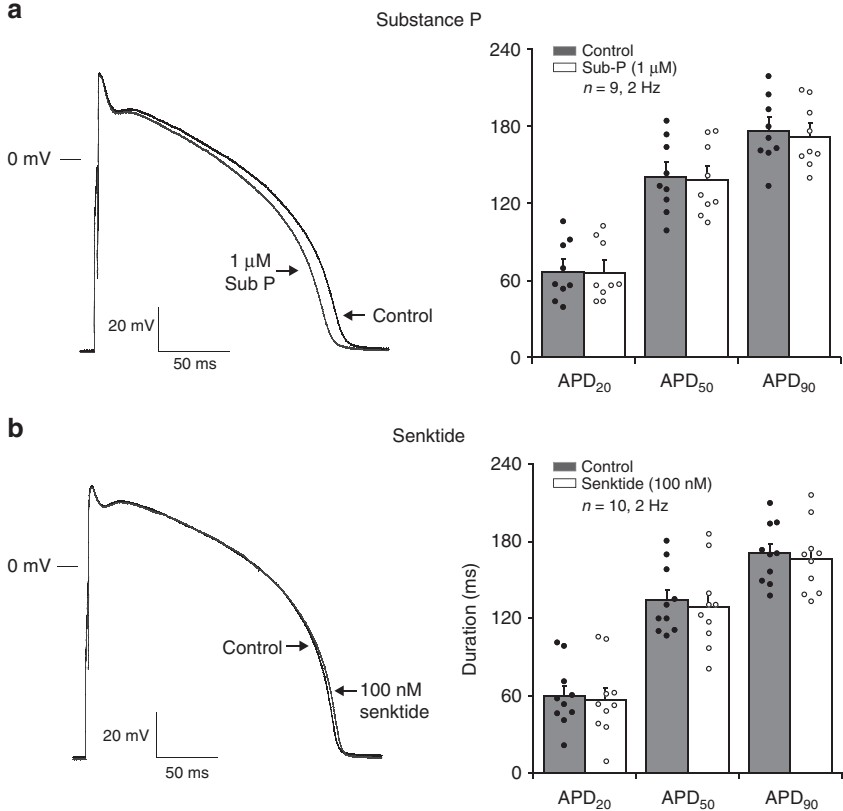

**Fig. 7** Effect of NK-3 receptor stimulation on action potentials of ventricular myocytes. (**a–b**, left panel) Ventricular APs elicited at 2 Hz under control conditions (Control) and in the presence of 1 μM Sub-P **a** or 100 nM Senktide **b**. (**a–b**, right panel) Bar graph showing average values for AP duration at 20, 50 and 90% of repolarization (APD$_{20}$, APD$_{50}$ and APD$_{90}$) before (Control) and after application of 1 μM Sub-P (**a**, $N = 5$, $n = 9$) or 100 nM Senktide (**b**, $N = 5$, $n = 10$). Paired $t$-test. All values shown are mean ± SEM. *$P < 0.05$

the electrophysiological actions of Sub-P on atrial cardiomyocytes are receptor-mediated involving the neurokinin cell-surface receptors[15,16]. We established that the AP prolongation by Sub-P was mediated through neurokinin-3 receptor (NK-3R) stimulation, as the selective NK-3R agonist Senktide[32] increased APD$_{90}$ with a 100 times higher potency than Sub-P, while NK-1R and NK-2R agonists at concentrations that maximally activate their respective receptors[33,34], did not change AP duration. Moreover, the AP prolongation by NK-3R stimulation was antagonized by the specific NK-3R antagonist Osanetant[35]. The presence of NK-3R on the surface of atrial cardiomyocytes has thus far not been demonstrated[18]. Hoover and Hancock[36] were unable to detect Sub-P binding sites in atria and ventricle of guinea-pig heart. Also Walsh et al.[37] did not find evidence for Sub-P binding sites in rat heart. However, neonatal rat ventricular cardiomyocytes express mRNA for NK-1R and NK-3R[38]. RNA-sequencing data and immunohistochemical analysis in our study further supports the presence of NK-3R in the sarcolemmal membrane of rabbit and human atrial cardiomyocytes. Thus, this demonstrates the existence of NK-3R on adult cardiomyocytes from rabbit and human atrium and to provide evidence of their functional significance. The RNA sequencing data indicate that the NK-3 receptor is the dominant neurokinin receptor in atrial cardiomyocytes as mRNA levels of NK-1R and NK-2R were relatively low. The endogenous ligand for the NK-3R is neurokinin B (NKB). However, neurokinin A (NKA) and Sub-P, as confirmed in the present study, also bind to the NK-3R, albeit with different affinities.[15,16] All three neurokinins are produced by neurons of the intrinsic cardiac nervous system.[19,39] The preferential affinity of NK-3R for the endogenous ligands has a potency order of NKB>NKA>Sub-P,

which implies that NKB may be more relevant for atrial electrophysiology than Sub-P. The exact nature of the effects of NKB/Sub-P on in vivo atrial electrophysiology are difficult to assess, as matters are further complicated by the fact that also (cholinergic) intracardiac neurons possess NK receptors and may release ACh upon stimulation.[40]

To our knowledge no data are available concerning the actions of Sub-P or Senktide on membrane currents in atrial cardiomyocytes. The main finding of the present study concerned the reduction in steady-state outward current in the presence of Sub-P and Senktide, which is consistent with the observed AP prolongation. Other studies have shown that a reduction in sustained outward current of similar amplitude, has important implications for atrial AP duration[41,42] Changes in outward I$_{Ca(Cl)}$, I$_{NCX}$ and I$_{to}$ do not contribute to AP prolongation, as their current amplitudes were unchanged in the presence of Sub-P, despite minor kinetic alterations in the latter. The reduction of I$_{Ca,L}$ is not compatible with AP prolongation by NK-3R stimulation. On the contrary, a decrease in inward current would shorten the AP. Evidently, the reduction in steady-state outward current by far outweighs the effect of I$_{Ca,L}$ reduction on AP duration. Detailed examination of the Sub-P sensitive current indicated that an outward K$^+$ current is involved, since its reversal potential is close to the calculated Nernst potential for K$^+$ ions. The steady-state outward current of atrial myocytes is composed of different types of K$^+$ currents, including the sustained component of the transient outward current (I$_{to}$), and the slow, rapid and ultra-rapid components of the delayed rectifier K$^+$ current (I$_{Ks}$, I$_{Kr}$ and I$_{Kur}$, respectively)[12,13]. Our experiments exclude a decrease in any of these current types as a possible mechanism underlying the

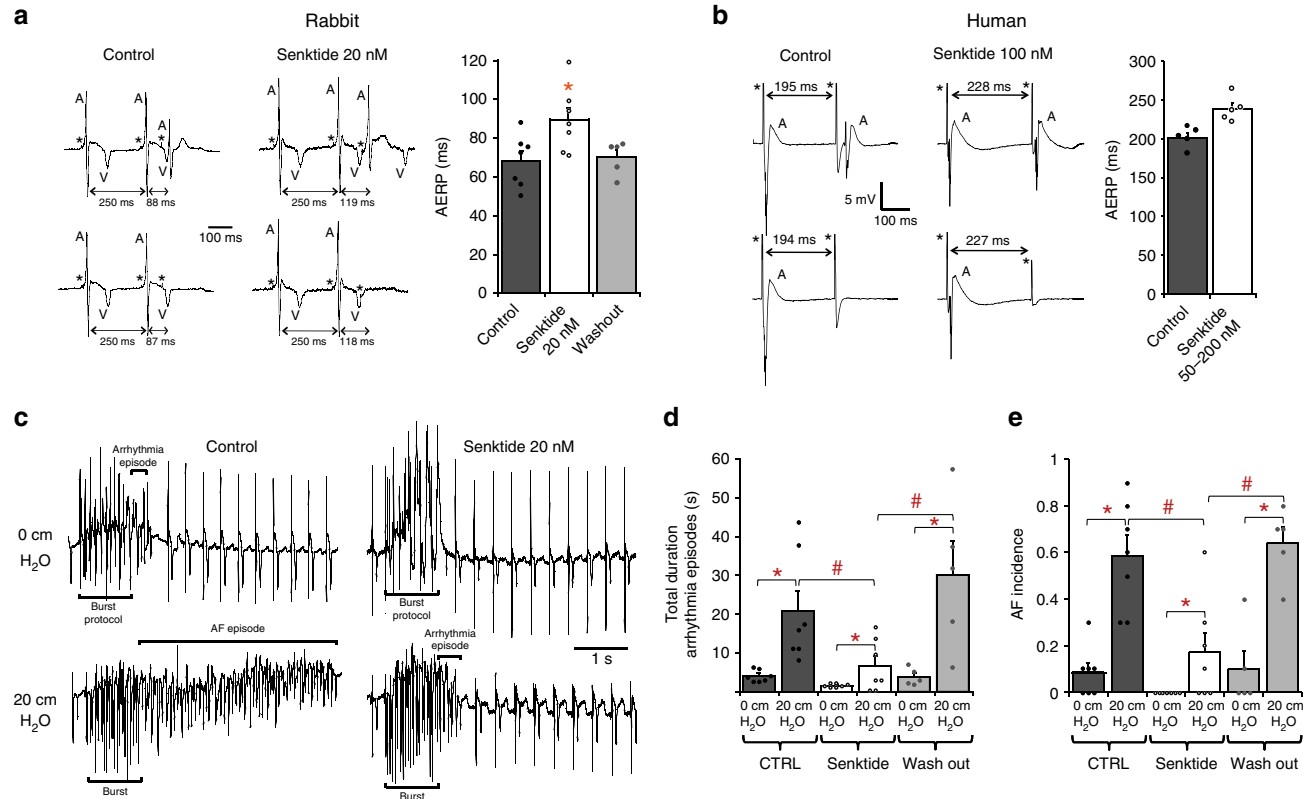

**Fig. 8** Effect of NK-3R stimulation on ERP and AF inducibility in atria of human and rabbit. **a** Unipolar electrograms recorded from the left atrium of a Langendorff-perfused rabbit heart before (Control) and after application of 20 nM of Senktide (left). Bar graph showing average values for AERP in Control, in the presence of 20 nM Senktide, and after wash-out (right, $N = 7$, one-way RM ANOVA). **b** Unipolar electrograms recorded from human LAA before (Control) and after application of 100 nM of Senktide (left). Bar graph showing average values for AERP in Control and in the presence of 50–200 nmol/L Senktide (right, $N = 5$, paired $t$-test, $p = 0.06$). **c** Unipolar electrograms recorded from the left atrium of a rabbit Langendorff-perfused heart following a burst pacing protocol in the absence (Control, left) and in the presence of 20 nM of the NK-3R agonist Senktide (right), at baseline (0 cm $H_2O$, upper tracings) and during atrial dilatation (20 cm $H_2O$, lower tracings). **d**, **e** Bar graph showing average values for total duration of arrhythmia episodes **d** and AF incidence **e** in Control, after application of 20 nM Senktide, and after wash-out ($N = 7$, two-way RM ANOVA). All values shown are mean ± SEM. *$P < 0.05$. Electrograms: A atrial activation; V ventricular activation; * stimulus

observed effects of Sub-P, since the reduction in steady-state outward current persists in the presence of the respective blockers. A candidate underlying the Sub-P mediated effect, is the background or leakage $K^+$ current, which was also inhibited by Sub-P in neurons[43–45]. Recent evidence suggests that members of the two-pore-domain $K^+$ (K2P) channel family generate the $K^+$ background conductance in excitable tissues[21]. Neurokinin receptors are known to be coupled to the Gq/G11 subgroup of G-proteins[22] and several members of the K2P family have been shown to be potently inhibited by receptors that signal through Gq/11, including TREK-1, TREK-2, TASK-1 and TASK-3[23,25]. As TASK-1, TASK-3 and TREK-1 channels have been functionally identified in atrial cardiomyocytes to contribute to AP duration[26–28], we considered these channels as candidates for the action potential prolonging effect. Nevertheless, selective inhibition of these channels scarcely affected AP duration, which makes their involvement in isolation unlikely. Application of the non-specific channel blocker $ZnCl_2$[42,46] abolished the inhibition of the background $K^+$ current by Senktide, and revealed the presence of a $ZnCl_2$-sensitive component with electrophysiological characteristics comparable to the Sub-P-sensitive and Senktide-sensitive currents. However, as zinc modulates the activity of a broad range of ion channels, the nature of this background $K^+$ current remains unsure. Future studies dedicated to exact channel identification, are required to designate a particular $K^+$ channel type.

The cardiac autonomic nervous system plays a significant role in the genesis and maintenance of AF. Particularly, alterations in the sympatho-vagal balance and excessive intrinsic cardiac nerve activity have been shown to trigger atrial arrhythmias[47,48]. The precise role of the various neuropeptides therein is still poorly understood. In this study, AP prolongation by NK-3R stimulation in isolated atrial cardiomyocytes (by Sub-P or Senktide) was reflected by an increased atrial effective refractory period (ERP) in rabbit Langendorff hearts and in situ anaesthetized rabbits, as well as in human isolated left atrial tissue. An increased ERP forms an anti-arrhythmic mechanism which may prevent or terminate AF[12,13]. To date however, pharmacological intervention aimed at lengthening of atrial repolarization has a limited applicability due to pro-arrhythmic ventricular side effects and reverse rate-dependence (i.e., a loss of action at high heart rates)[12,13]. As NK-3R stimulation does not prolong the ventricular AP, and prolongs the atrial AP also at high heart rates, it meets the important prerequisites for a potential atrial-specific, anti-fibrillatory action. Indeed we show that NK-3R stimulation by Senktide in a rabbit isolated heart model of AF based on atrial dilatation, reduces AF incidence and duration by 70%. Thus, we demonstrate that the electrophysiological effects of NK-3R stimulation are of potent anti-fibrillatory nature.

In conclusion, this study provides evidence for the functional expression of the neurokinin-3 receptor in the sarcolemma of atrial cardiomyocytes. This receptor was known to be expressed

by neural tissue, but was hitherto unidentified in cardiac atrial muscle. Stimulation of NK-3R produces significant prolongation of the atrial but not the ventricular AP through inhibition of a background $K^+$ channel, possibly a member of the K2P family. The AP-prolonging action of NK-3R stimulation in the atrium was also present at high heart rates and greatly reduced AF inducibility. Hence, we propose NK-3R as a potential anti-arrhythmic drug target for treatment of AF.

## Methods

**Animals and human tissue.** The study conformed to the 'Guide for the Care and Use of Laboratory Animals' published by the US National Institutes of Health (NIH Publication No. 85–23, revised 1996) and was approved by the local animal experiments committee of the Academic Medical Center, Amsterdam, The Netherlands. The study on human atrial appendages was in accordance with the declaration of Helsinki and was approved by the Institutional Review Board of the Academic Medical Center, Amsterdam, The Netherlands. All patients gave written informed consent.

**The Langendorff-perfused rabbit heart.** Male New Zealand White rabbits (2.5–3.5 kg) were anaesthetized with 20 mg xylazine and 100 mg ketamine (intramuscularly) and heparinized with a bolus of 1000 IU heparin (intravenously). Subsequently, the animals were killed by 200 mg pentobarbital intravenously, the thorax was opened and the heart was quickly removed and submerged in ice-cold perfusion solution (composition see below). After cannulation of the aorta, the heart was mounted on a Langendorff perfusion setup, and perfused at 37 °C with a modified Tyrode's solution containing (in mmol/L) 128 NaCl, 4.7 KCl, 1.45 $CaCl_2$, 0.6 $MgCl_2$, 27 $NaHCO_3$, 0.4 $NaH_2PO_4$ and 11 glucose (pH maintained at 7.4 by equilibration with a mixture of 95% $O_2$ and 5% $CO_2$)[49].

**Atrial refractoriness in the Langendorff rabbit heart.** Bipolar stimulation- and unipolar recording electrodes were placed on the left atrial myocardium. The heart was stimulated at a basic cycle length of 200 ms (cathodal stimulation, 1 ms current pulse duration, twice the diastolic stimulation threshold). After an equilibrium period of 30 min, the effective refractory period (ERP) was determined by a train of 16 stimuli at 200 ms followed by a single stimulus with progressively shortened coupling interval until loss of capture. The ERP was defined as the longest S1–S2 interval without capture. After an equilibrium period of 30 min, wash-in of 20 nM NK-3R-agonist Senktide was started and ERP measurements were repeated 10 min following wash-in.

**Rabbit isolated heart model of AF based on atrial dilatation.** After mounting of the heart on the Langendorff perfusion setup, a distendable balloon (volume 4–5 ml) connected to a manometer, was inserted through the pulmonary vein openings into the left atrium, Subsequently, ±0.5 ml saline was injected into the balloon to yield an intra-atrial pressure of 0 cm $H_2O$ (baseline condition). A bipolar pacing electrode was positioned at the right atrium free wall and the heart was stimulated at a basic cycle length of 250 ms. Unipolar recording electrodes were positioned on the left atrium free wall and the left ventricle wall to record atrial and ventricular electrograms, respectively. The heart was then allowed to equilibrate for a 20-min period of continuous stimulation, after which right atrial effective refractory period (AERP) was determined by a train of 8 stimuli at 250 ms followed by a single stimulus with progressively shortened coupling interval until loss of capture. The ERP was defined as the longest S1–S2 interval without capture. Subsequently, after 20 stimuli at the basic cycle length (250 ms), a burst pacing protocol consisting of 20 stimuli at 20 Hz (2 ms current pulse duration) was delivered to provoke AF, followed by a 5 s pause. The protocol to evoke AF was repeated ten times. Next, the left atrial balloon was distended by injection of 2–4 ml saline up to an intra-atrial pressure of 20 cm $H_2O$ to dilate the atrium, and AERP measurements and burst pacing protocol to provoke AF were repeated. After a 5 min. recovery period at 0 cm $H_2O$, drug infusion was started and the complete procedure was repeated in the presence of 20 nM Senktide. Then, a 30 min wash-out period (at intra-atrial pressure 0 cm $H_2O$) followed, and AERP and AF inducibility were determined again at intra-atrial pressures of 0 and 20 cm $H_2O$, respectively.

For data analysis we considered three groups: (1) no arrhythmia, (2) an arrhythmia episode consisting of short runs of premature atrial activations with a duration $\Delta t$ <1 s, and (3) an irregular arrhythmia episode with $\Delta t$ >1 s in duration, which was defined as AF. To assess the anti-arrhythmic effect of Senktide we determined the total duration of arrhythmias episodes, which was defined as the sum of durations of arrhythmia episodes following 10 consecutive burst pacing protocols, and the AF incidence, which was defined as the fraction of the 10 consecutive burst pacing protocols imposed, that evoked AF.

**Left atrial refractoriness in the anaesthetized rabbit.** Male New Zealand White rabbits (2.5–3.5 kg) were anaesthetized with ketamine 50 mg/kg and xylazine 8 mg/kg, and received 0.09 mg/kg Temgesic subcutaneously. The rabbits were intubated and artificially ventilated. Anaesthesia was maintained by isoflurane

(0.8–1.2%). Catheters for drug infusions and blood pressure recording were inserted into marginal ear veins and arteries, respectively. Following thoracotomy, the pericardium was opened and the heart was instrumented as above. When the rabbit was hemodynamically and electrophysiologically stable, the atrial ERP was determined by a train of 8 stimuli at 220 ms followed by a premature stimulus with progressively shortened coupling interval until loss of capture. Subsequently, a single bolus injection (1 ml i.v.) of 11 nmol/kg Senktide was administered and the measurement protocol was repeated.

**Refractory periods in human left atrial appendages.** Left atrial appendages (LAA) were routinely removed from patients with atrial fibrillation undergoing minimally invasive surgical pulmonary vein isolation with an endoscopic stapling device (Endo Gia stapler, Tyco Healthcare Group)[50]. The tissue samples were transported in 100-ml cooled solution I (see modified Tyrode's solution above) containing 1000 IE heparin, and submerged in a tissue bath. The superfusion fluid was kept at a stable temperature of 36.5–37.5 °C and gassed with a mixture of 95% $O_2$ and 5% $CO_2$ (pH 7.4). LAAs were stimulated at 100 beats per minute at 2–3 times diastolic threshold with a pulse width of 2 ms using a bipolar epicardial electrode.

The ERP was determined by a train of 8 stimuli (at 600 ms interval) followed by a single stimulus with progressively shortened coupling interval until loss of capture. After an equilibrium period of 30 min, Senktide superperfusion (50–200 nM) was started, and the ERP protocol was repeated.

**Single cardiomyocyte preparation.** After excision of the heart from male New Zealand White rabbits[49], single left atrial and ventricular cells were isolated as described previously, with minor modifications[51,52]. In short, hearts were mounted on a Langendorff perfusion apparatus and retrogradely perfused at 37 °C through the aorta with a modified Tyrode's solution (see above) at a constant pressure (50 mmHg) for 15 min. Next, perfusion (50 mmHg) was changed to a nominally calcium-free dissociation solution containing in mM: HEPES 16.8, NaCl 146.5, KHCO_3 3.3, KH_2PO_4 1.4, NaHCO_3 1.0, MgCl_2 2.0, CaCl_2 0.01, glucose 11.0, pH 7.3 (NaOH). After 15 min, collagenase type B (0.15 mg/ml, Roche 11088815), collagenase type P (0.05 mg/ml, Roche 11213865), trypsin inhibitor (0.1 mg/ml, Roche 10109878), 0.2 mg/ml hyaluronidase (Sigma H-3506), protease XIV (Sigma H-3506) and creatine (10 mM) were added. During this last period, the heart was perfused at a constant flow in a recirculating manner. When perfusion pressure dropped from an initial value of 50 to <2 mmHg (usually after about 30 min), the left ventricular wall and the left atrium were cut into small pieces and further fractionated in the enzyme-containing dissociation solution using a gyrotory water bath shaker (25 min). During the last 10 min, 1% albumin (fatty acid free, Roche 10775835001) was added to the enzyme-containing dissociation solution. All dissociation solutions were saturated with 100% $O_2$ and the temperature was maintained at 37 °C. Cells were allowed to sediment and were resuspended in (enzyme-free) dissociation solution to which 1% albumin and 1.3 mM $CaCl_2$ was added.

Small aliquots of single cell suspension were introduced into a recording chamber on the stage of an inverted microscope. Cells were allowed to adhere for 5 min after which superfusion was started. Single quiescent rod-shaped myocytes with clear cross-striations and smooth surfaces were selected for measurements.

**Cellular electrophysiology.** Action potentials (APs) and membrane currents were recorded at 36.5 °C with the amphotericin-B-perforated or ruptured patch clamp technique[51], using an Axopatch 200B Clamp amplifier (Molecular Devices Corporation, Sunnyvale, CA, USA). Voltage control, data acquisition, and analysis were performed using custom-made software.

Series resistance was compensated for and potentials were corrected for liquid junction potential[53]. Signals were low-pass filtered (cutoff frequency: 5 kHz) and digitized at 40 kHz. Cell membrane capacitance ($C_m$) was estimated by dividing the time constant of the capacitive transient in response to 5 mV hyperpolarizing voltage clamp steps from a holding potential of −40 mV, by the series resistance.

**Current-clamp experiments.** For AP measurements, a standard superfusion solution was used containing (in mM): NaCl 140, KCl 5.4, $CaCl_2$ 1.8, $MgCl_2$ 1.0, glucose 5.5, HEPES 5.0, pH 7.4 (NaOH). The patch-pipettes (borosilicate glass; 1–3 MΩ) were filled with a standard pipette solution containing (in mM): K-gluconate 110, KCl 30, NaCl 5, $MgCl_2$ 1, amphotericin-B 0.22, HEPES 10, pH 7.3 (KOH).

APs were elicited at 1–4 Hz by 2-ms (1.5× diastolic stimulation threshold) current pulses applied through the patch pipette. Resting membrane potential (RMP), maximal upstroke velocity ($V_{max}$), AP amplitude (APA) and AP duration (APD) at 20%, 50% and 90% repolarization ($APD_{20}$, $APD_{50}$ and $APD_{90}$, respectively), were obtained from 10 consecutive APs and averaged.

Cardiomyocytes were allowed to equilibrate for a 5-min period of continuous stimulation (1 Hz or 2 Hz) after which Sub-P (1 μM) or one of the specific neurokinin receptor agonists (100 nM) was administered. Agonists for the NK-1R, NK-2R and NK-3R were respectively: [Sar9,Met(O2)11]-SP[33], (β-Ala)-NKA(4-10)[34] and Senktide[32]. APs were recorded just before, and 3–5 min after application of the neuropeptides. In dose-response experiments, Sub-P or Senktide was applied in a cumulative sequence with 5 min intervals between increments in concentration

(Sub-P: 1, 10, 100 nM, 1, 10 µM; Senktide: 1, 2, 5 nM and 10, 20, 50 nM). The concentration of 100 nM Senktide was applied as a singular dose.

Cardiomyocytes were pre-incubated with the specific NK-3R antagonist Osanetant[35] ((R)-N-{{3-[1-Benzoyl-3-(3,4-dichlorophenyl)piperidin-3-yl]prop-1-yl]-4-phenylpiperidin-4-yl]-N-methylacetamine) (300 nM) for 10 min to study the specificity of the Senktide-mediated APD prolongation.

To assess the contribution of TASK-1, TASK-3 and TREK-1 currents to the action potential, $APD_{90}$ was measured after application of the selective blockers ML-365[54] (100 nM), PK-THPP (100 nM)[55] and Spadin[56] (100 nM), respectively. APs were recorded just before, and 3–5 min after application of the channel blockers.

**Voltage-clamp experiments.** Steady-state currents, L-type $Ca^{2+}$ current ($I_{Ca,L}$), transient outward $K^+$ current ($I_{to}$), $Ca^{2+}$-activated $Cl^-$ current ($I_{Cl(Ca)}$) and $Na^+$–$Ca^{2+}$ exchange current ($I_{NCX}$) were measured with solutions specified below and using voltage-clamp protocols depicted in the corresponding figures. Steady-state currents (current at the end of the 500-ms voltage step), $I_{to}$ and $I_{Cl(Ca)}$ were measured using the amphotericin-B-perforated patch clamp technique with standard pipette- and superfusion-solution (see current-clamp experiments above). To accurately determine $I_{to}$, measurements were performed in the presence of blockers of $Na^+$ channels (30 µM tetrodotoxin (TTX)), $Ca^{2+}$ channels (0.25 mM $CdCl_2$) and of delayed rectifier $K^+$ channels (5 µM E-4031, 100 µM chromanol 293B).[51] For $I_{Cl(Ca)}$ measurements, 10 µM TTX and 2 mM 4-aminopyridine (4-AP) was added to the standard superfusion solution.[57]

$I_{Ca,L}$ and $I_{NCX}$ were measured using the ruptured patch clamp technique. $I_{Ca,L}$[57] was measured with pipette solution containing (in mM): CsCl 145, $K_2$-ATP 5, EGTA 10, HEPES 10, pH 7.2 (NMDG-OH). The superfusion solution contained (mM): TEA-Cl 145, CsCl 5.4, $CaCl_2$ 1.8, $MgCl_2$ 1.0, glucose 5.5, HEPES 5.0; pH 7.4 (NMDG-OH). $I_{Ca}$ was measured in the presence of 0.25 mM 4,4'diisothiocyanatostilbene-2,2'-disulfonic acid (DIDS; Sigma-Aldrich, MO, USA) to block $I_{Cl(Ca)}$.[57] $I_{NCX}$[58] was measured with pipette solution containing (in mM): CsCl 145, NaCl 5, Mg-ATP 10, TEA-Cl 10, EGTA 20, $CaCl_2$ 10, HEPES 10, pH 7.2 (NMDG-OH). The superfusion solution consisted of a $K^+$-free solution to which 1 mM $BaCl_2$, 2 mM CsCl, 5 µM nifedipine, 100 µM ouabain and 200 µM DIDS was added to suppress membrane currents other than $I_{NCX}$. $I_{NCX}$[58] was measured as 10 mM $Ni^{2+}$-sensitive current during a descending voltage ramp. As the effects of $Ni^{2+}$ on $I_{NCX}$ are reversible,[58] $I_{NCX}$ measurements in the absence and presence of Sub-P analogues were carried out in the same cell.

In voltage clamp experiments the effect of Sub-P and Senktide on the various membrane currents was assessed at a concentration of 10 µM and 50 nM, respectively, 3–5 min after application.

To test for the functional presence of members of the K2P family, steady-state currents were also measured after application of the ion channel blocker $ZnCl_2$ (100 µM).[29–31]

Voltage dependencies of (in)activation were determined by fitting a Boltzmann function ($y = A/[1 + \exp\{(V-V_{1/2})/k\}]$) to the individual curves, yielding a half-maximal voltage $V_{1/2}$ (mV) and a slope factor $k$ (mV). Time constants of inactivation were obtained by fitting current decay with a bi-exponential function $y = y_0 + A_f\exp(-t/\tau_f) + A_s\exp(-t/\tau_s)$, where $A_f$ and $A_s$ are the amplitudes of the fast and slow inactivating components, and $\tau_f$ and $\tau_s$ their respective inactivation time constants. Current densities were calculated by dividing the current amplitude by $C_m$.

**Cytosolic $Ca^{2+}$ transients.** Intracellular $Ca^{2+}$ ($Ca^{2+}_i$) was measured in indo-1-am loaded atrial cardiomyocytes as described previously.[59] Dual wavelength emission of indo-1 was recorded ((405–440)/(505–540) nm, excitation at 340 nm) and free $Ca^{2+}_i$ was calculated. $Ca^{2+}_i$ transients were elicited at 5 Hz using field stimulation. For determination of diastolic and systolic $Ca^{2+}_i$ concentrations, and $Ca^{2+}_i$ transient amplitudes, data from 10 consecutive $Ca^{2+}_i$ transients were averaged. The effect of Sub-P on $Ca^{2+}_i$ was assessed at a concentration of 10 µmol/L, 3 min after application.

**Immunohistochemistry.** Rabbit left atrium and human left atrial appendages were snap-frozen in liquid nitrogen, and stored at −80 °C. Cryosections (7 µm) were mounted on 3-aminopropyltriethoxysilane (AAS)-coated glass slides and permeabilized in 0.2% Triton X-100 in PBS for 20 min, where after they were blocked in 2% bovine serum albumin for 30 min. Human cryosections were incubated overnight with the following primary antibodies: polyclonal anti-NK3R (Immunostar 20061; 1:50 dilution) and monoclonal anti-α-actinin (Sigma T7811; 1:1000). Next, they were incubated for 90 min with secondary antibodies Alexa-conjugated goat anti-mouse and goat anti-rabbit antibodies (1:250, Molecular Probes, Invitrogen) in 10% Normal Goat Serum (at room temperature). For double labelling, sections were incubated with a mixture of primary antibodies, followed by an appropriate mixture of secondary antibodies.[60] Rabbit cryosections were incubated overnight with the polyclonal primary antibody anti-NK3R (Abcam ab123303, 1:5 dilution) directly labelled with ATTO-488 fluorophore using Lightning-Link® Rapid Conjugation System (Innova Biosciences, Cambridge, UK), to eliminate background staining on rabbit tissue. Confocal imaging was performed using a confocal laser scanning microscope (BioRad MRC1024) equipped with a 15-mV Krypton/Argon laser, using the 568 and 488 excitation lines and 605DF32 and 522DF35 emission filters.

**RNA-sequencing.** After excision of the heart from a male New Zealand White rabbit, tissue samples from the left atrial free wall and left atrial appendage were isolated (4 mm³) and snap frozen in liquid nitrogen. RNA was isolated using TruSeq (Illumina) and sequenced using Illumina Hiseq 4000 (sequence length 50, single end). Reads are subjected to quality control (FastQC, Picard Tools), trimmed using Trimmomatic v0.32[61] and aligned to the genomes using HISAT2 (v2.0.4)[62]. The genome and GTF were obtained from Ensembl (v0.89): OryCun2.0 (assembly GCA_000003625.1) and Oryctolagus_cuniculus.OryCun2.0.89.asFasta.gtf, respectively. Counts are obtained using HTSeq (v0.6.1)[63]. Statistical analyses are performed using the edgeR[64] and limma/voom[65] R packages. All genes with no counts in any of the samples are removed whilst genes with more than five reads in at least two of the samples are kept. Count data are transformed to log2-counts per million (logCPM), normalized by applying the trimmed mean of M-values method[64] and precision weighted using voom[66]. Differential expression is assessed using an empirical Bayes moderated $t$-test within limma's linear model framework, including the precision weights estimated by voom. Resulting $P$-values are corrected for multiple testing using the Benjamini-Hochberg false discovery rate. Genes are re-annotated using biomaRt using the Ensembl genome databases (v91). For ~65% of Ensembl IDs an Entrez Gene ID was retrieved using biomaRt..

**Drugs.** All drugs were obtained from Sigma-Aldrich (MO, USA), except for E-4031, PK-THPP, ML-365 (Tocris, MN, USA), Sub-P (Enzo Life Sciences, NY, USA), and TTX (Abcam Biomedicals, Cambridge, UK).

DIDS was freshly prepared as a 0.5 M and chromanol 293B as a 0.1 M stock solution in DMSO. Nifedipine was prepared as a 5 mM stock solution in ethanol. E-4031 and TTX were prepared as a 5 and 30 mM stock solution in distilled water. Senktide was prepared as a 1 mM stock solution in ethanol. PK-THPP and ML-365 were prepared as a 1 mM stock solutions in DMSO. Spadin was prepared as a 1 mM Stock solution in distilled water. $ZnCl_2$ was prepared as a 100 mM stock solution in distilled water. All stock solutions were diluted appropriately before use. Sub-P was freshly dissolved at its final concentration. DIDS and nifedipine were stored in the dark.

**Statistics.** The data are presented as mean ± SEM. Normality and equal variance assumptions were tested with the Kolmogorov–Smirnov and the Levene median test, respectively. Groups were compared using a paired Student's $t$-test (two-sided), one-way repeated measures (RM) analysis of variance (ANOVA), or two-way RM ANOVA followed by post hoc comparison using the Student–Newman–Keuls method, where appropriate. Factor correction was applied on APD (in Figs. 1e and 4d) and ERP (in Fig. 8 and Supplementary Figure 5) using day of experiment and the experimental animal as session factor, respectively[67]. N is number of hearts, n is number of cells. $P < 0.05$ was defined as statistically significant.

## Data availability

The data supporting the findings of this manuscript are available from the corresponding authors upon reasonable request. RNA sequencing data that support the findings of this study have been deposited in Geo with the accession code GSE117801.

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

## Acknowledgements

R.C. is supported by the Leducq International network of Excellence RHYTHM. J.R.G. is supported by a personal VIDI grant from the Netherlands Organization for Scientific Research NWO/ZonMW 016.146.310. C.A.R. is supported by a personal VIDI grant from the Netherlands Organization for Scientific Research NWO/ZonMW (91714371). B.J.B. is supported by a personal grant from the Dutch Heart Foundation (2016T047).

## Author contributions

M.W.V. conceived the project, designed and performed the experiments, analysed and interpreted the data, and wrote the manuscript. G.S.C. performed the experiments, analysed and interpreted the data and wrote the manuscript. A.B. performed the experiments, analysed and interpreted the data and wrote the manuscript. A.O.V. performed the experiments, analysed and interpreted the data and wrote the manuscript. C. A.S. performed the experiments and analysed and interpreted the data. G.G.S. performed the experiments and analysed the data. W.R.B. performed the experiments and analysed the data. S.C. performed the experiments and analysed the data. S.C.M.A. performed the experiments and analysed the data. K.T.S. performed the experiments and analysed the data. A.H.G.D. interpreted the data and edited the manuscript. C.N.W.B. performed the experiments. A.C.G.G. interpreted the data and edited the manuscript. J.R.G. interpreted the data and edited the manuscript. J.M.T.B. interpreted the data and edited the manuscript. C.A.R. interpreted the data and wrote the manuscript. B.J.B.

performed the experiments and the statistical analysis, and edited the manuscript. R.C. conceived the project, designed the experiments, interpreted the data and wrote the manuscript.

## Additional information

**Competing interests:** The authors declare no competing interests.

