## [Peer Review File · Nature Communications]

Reviewers' comments:

Reviewer #1 (Remarks to the Author):

The study by Veldkamp et al describes the effects of Neurokinin-3 receptor stimulation on atrial action potentials and excitability. Besides the known role of the cardiac nervous system on atrial fibrillation and that endogenous ligands like substance P might be involved, the authors now claim that stimulating receptors of this nervous system might be beneficial for the treatment of atrial fibrillation. Using patch clamp experiments under stimulation of the neurokinin receptors the authors claim that the antiarrhythmic mechanism of action is a reduced conductance of the K_{2P} potassium channel TASK-3. The main problem of the manuscript is the insufficient explanation of the effects seen by neurokinin receptor stimulation. The effects on background conductance was given with approximately 25 % and a just weakly significant effect $p < 0.02$ (end of page 5). The rectification of the difference current provided in figure 2A (III) is not typical for a K_{2P} channel (no outward rectification present and therefore an increased inward conductance at -120 mV). In addition, there is no evidence that TASK-3 is involved. First, the functional expression of TASK-3 in rabbit atrium is not investigated at all by molecular biological methods. Second, the drugs the authors use (anandamide and Zn²⁺) to claim an involvement of TASK-3 are definitively NOT specific blockers of a particular type of ion channel, although the authors claim this in an attempt to try to explain the observed effects. There is not enough evidence to claim an involvement of any particular conductance in the effects observed with substance P.

Reviewer #2 (Remarks to the Author):

This paper is clearly presented and makes a good case (based on the use of inhibitors of current receptors), that Substance P activates a voltage-independent K channel in atrial but not ventricular myocytes, and that this prolongs action potential duration (APD) in rabbit and human atria. Moreover, they show evidence that this requires the NK-3 receptor and a 2-pore K channel (TASK-3). I have some concerns about the complete reliance on inhibitors, with little discussion about selectivity (or lack thereof) The authors should be more forthcoming on this issue.

Another concern is that the links between these novel observations about the role of the NK-3 receptor in atrial APD prolongation and the intrinsic cardiac nervous system (ICNS) as stated in the Abstract and Introduction) are weak here. It seems more prudent for the authors not to muddy the waters by talking so much about the ICNS. This point could be mentioned in the Discussion rather than abstract/intro), but the text should focus on the fact that the NK-3 receptor may be a novel target for treatment/prevention of AF.

A related point is that the authors should directly discuss interaction of this substance P effect with other autonomic neurotransmitters that may be locally co-secreted with substance P in the heart. Indeed, a few experimental tests may be worthwhile. Substance-P is released by nerves in the ICNS, but Sub-P has highest affinity for NK-1 receptors (located on neurons and other cell types) and lowest affinity for NK-3 receptors (myocytes). So at physiological Sub-P levels (which may not be known in the heart), the predominant action of Sub-P may be on neurons, where it stimulates cholinergic neurons, and which would likely produce atrial APD shortening, not prolongation. I'm pretty sure this is why Sub-P was used in the manuscript to study isolated atrial myocytes, but a specific NK-3 receptor agonist was used in the whole-heart and in vivo experiments, leaving this question open (maybe that's why they used the selective agonist). Therefore, relating these findings to physiological function of the ICNS is a little bit of a stretch as it stands. This doesn't detract from the importance of the findings, though, with respect to the NK-3 receptor specifically. Just maybe influences how they should tell the story.

Other points:

Page 4. The 6 mV depolarization of resting E_m with Sub-P is surprising and begs the question as to how much IK1 vs. Sub-P sensitive K current controls diastolic E_m and stability with respect to triggered arrhythmias like DADs in atrium (vs. ventricle). It is also a bit surprising that the 6 mV depolarization did not reduce Na channel availability significantly (as in V_{max}) and conduction. Pg 4 Line n-4 " a background K-selective channel current."

The effectiveness and selectivity of specific ion-channel block was not verified in general. This may be less important for the broad conclusion from Fig 2, but seems missing for Fig 3, where the Ca, NCX and Cl(Ca) currents reported should be shown to be sensitive to the inhibitors used elsewhere (e.g. Cd or nifedipine for Ca current, Ni for NCX and DIDS for Cl(Ca)). This concern is especially critical for the use of 100 μ M Zn as a specific TASK-3 inhibitor (later in Fig 7). That concentration of Zn is expected to influence many things.

Pge 7, last full paragraph. Are you implying that Sub-P is working via Gs-alpha here? Via cAMP and PKA? Or is the NK-3 receptor coupled to TASK-1 & 3 via Gq11 as mentioned in the Discussion? Try to be clearer about what is known/expected and hypothesized.

Fig 8. Is there any immunofluorescence control data to show that the NK3R antibody is specific and not picking up some other protein (a notorious problem for many antibodies –especially for sparsely expressed channels and receptors)? Even showing parallel ventricular myocyte data as a negative control would be somewhat reassuring.

Discussion, pg. 19: "On the contrary, a decrease in inward current would lengthen the AP." I think you mean shorten the AP....?

Figure 7 was missing, but I got it before submitting this review.

Reviewer #3 (Remarks to the Author):

The paper "NEUROKININ-3 RECEPTOR ACTIVATION SELECTIVELY PROLONGS ATRIAL REFRACTORINESS BY INHIBITION OF A BACKGROUND K⁺ CHANNEL" submitted by Veldkamp et al. concerns elucidation of a potential new neurokinin pathway potentially important in the pathophysiology of atrial fibrillation. The authors nicely show (at 36.5 C – very nice) a number of electrophysiological experiments on isolated atrial myocytes. These experiments reveal that the APD prolonging effect of Sub-P is probably through an inward rectifier channel – presumably TASK-3. Also, the authors show that the effect of Sub-P is probably via NK-3R. This all seems well funded, although I have one concern (see below) in that respect.

The results revealed in this paper are truly new and interesting. They should have a large interest in the cardiac electrophysiological community as papers like this help us to bridge the gap in knowledge between cardiomyocyte electrophysiology and neuronal communication/interference/stimulation. However, to reach the standards of Nature Communications I believe the following issues need to be addressed:

Major concerns:

(Patho-)physiology. What I really miss is evidence for the involvement of Sub-P in atrial (patho-)physiology. This could relatively easily be achieved with the Langendorff set-up of isolated rabbit hearts. By tachypacing, the vulnerability to AF could be evaluated. This should be done both in the presence of agonists and antagonists. Langendorff heart perfused with saline buffer will be slightly ischemic and as this might trigger Sub-P release application of an antagonist would be of interest. Also, application of Sub-P, and not only Senktide should be done. Another option is to use the thoracotomised rabbit model for these investigations.

Power. The number of experiments performed in most of the procedures are very low. In particular the n numbers in the ex vivo experiments are critically low. Please include power calculations/arguments or/and increase numbers.

Concentration of Senktide. For the rabbit Langendorff experiments you have used 20-50 nmol/l of

Senktide. I don't understand this. Why not the same concentration? And if not, can you at all group these results?

Time-matched controls. It seems like the electrophysiological experiments are performed in a paired fashion with control conditions followed by drug application. This is an acceptable way to perform the experiments, however, it is necessary to confirm that time from break-through/explantation is not a factor influencing the recordings. This has not been done.

Immunohistochemistry. The stainings reveal a nice membrane-like expression of NK-3R in rabbit and human atrial myocytes. The primary AB is used in a 1:20 dilution. This reviewer miss the controls. Please refer to other publications where this AB has been carefully evaluated, or conduct/show this experiments yourself.

Minor:

Line 77-83 could be condensed as they are more or less repeating information from abstract and other places in the MS.

Line 107-108: At which time-point after the step was this recorded?

Line 225. Has any compound ever been found to be specific? And a divalent ion? You should delete "specific".

Figure 1: You should show RMP and APA graphs. These are important when trying to understand the effect on AF vulnerability etc.

Figure 1-7: Time-matched controls – at least in some of the figures

Figure 6: Why do you use Senktide instead of Sup-P? This should be explained more carefully in the text.

Response to Reviewer #1

We thank the Reviewer for the stimulating comments and useful suggestions.

In response to the comments of the Reviewers, we have made substantial revisions to the manuscript and have performed additional experiments, including implementation of time controls, measurements with more specific blockers for TASK-1, TASK-3 and TREK-1, additional molecular analyses including RNA-sequencing, and assessment of the anti-arrhythmic action of Senktide in a rabbit isolated heart model of AF.

The Reviewer's comments are included in italic typeface, our response in regular typeface.

The main problem of the manuscript is the insufficient explanation of the effects seen by neurokinin receptor stimulation. The effects on background conductance was given with approximately 25 % and a just weakly significant effect $p < 0.02$ (end of page 5).

While it may not necessarily fully explain the effects of neurokinin receptor stimulation, the observed reduction in outward current of 25% in the range of the action potential plateau is functionally very relevant. The membrane resistance during the cardiac action potential plateau is relatively high, and small changes in net inward or outward current during this phase, can cause considerable changes in action potential duration. In our study we found that a reduction of ~1.5 pA/pF (20-44% outward current reduction; new Figure 2c (Sub-P), new Figure 6c (Senktide)) gives rise to an increase in APD₉₀ of 30-50 msec (30-50% APD₉₀ increase; new Figure 1e (Sub-P), new Figure 4d (Senktide)). This change in AP duration is biologically highly relevant. Moreover, these effect sizes are in line with observations from other studies. Schmidt et al. show that inhibition of TASK-1 (size: ~1.7 pA/pF) in human atrial myocytes gives rise to a 58% action potential (AP) prolongation (Schmidt et al. *Circulation* 2015;132:82–92;). In addition, mathematical modeling studies have shown the importance of sustained, non-inactivating K⁺ currents on atrial AP duration. (Nygren et al. *Circ Res.* 1998;82:63-81). Hence, we consider the observed decrease in background current of ~25% upon neurokinin-3 receptor stimulation (which was statistically, and not weakly, significant at $p < 0.02$) of significant functional relevance. We have now included a sentence in the Discussion session, stating that: "Other studies have shown that a reduction in sustained outward current of similar amplitude, has important implications for atrial AP duration" (pg 11, line 2).

The rectification of the difference current provided in figure 2A (III) is not typical or for a K2P channel (no outward rectification present and therefore an increased inward conductance at -120 mV).

As reviewed by Feliciangeli et al., not all members of the K2P family have the same electrophysiological properties. Although many of them have a linear conductance (e.g. TASK-1), others exhibit slight outward rectification (e.g. TREK1) or inward rectification (e.g. TWIK1, TWIK2) (Feliciangeli et al. *J Physiol.* 2015;593:2587–2603). TASK-3 potassium channels display outward rectification. Thus, the absence of outward rectification does not exclude the possibility the current being carried by K2P channels.

Indeed, as noted by the Reviewer, the Sub-P sensitive current in new Figure 2c has a more or less linear conductance (no outward rectification). On the other hand, specific activation of the NK-3 receptor by Senktide does result in inhibition of an outwardly-rectifying current (new new Figure 6c). This difference between Sub-P sensitive-current and Senktide sensitive-current, may be explained by the fact that Sub-P in addition to activating the NK-3 receptor, also activates

the NK-1 receptor, which may inhibit e.g. the inward rectifier channels (as shown in neurons by Shen K.Z. and North, R.A. Substance P opens cation channels and closes potassium channels in rat locus coeruleus neurons. 50, 345–353 (1992).

In addition, there is no evidence that TASK-3 is involved. First, the functional expression of TASK-3 in rabbit atrium is not investigated at all by molecular biological methods. Second, the drugs the authors use (anandamide and Zn²⁺) to claim an involvement of TASK-3 are definitively NOT specific blockers of a particular type of ion channel, although the authors claim this in an attempt to try to explain the observed effects. There is not enough evidence to claim an involvement of any particular conductance in the effects observed with substance P.

The point of the Reviewer is well taken. In the original manuscript, we did not claim (i.e. state as a fact) that TASK-3 is involved; instead, we attempted to use careful phrasing using words such as “suggesting” and “presumably” when referring to this point. Nevertheless, this issue may have still been unclear and insufficiently explained. Moreover, we have indeed incorrectly used the word “specific” blocker, which suggests that anandamide and Zn²⁺ solely inhibit TASK-1 and TASK-3 channels, respectively. Anandamide is a TASK-1 selective blocker in comparison with the other members of the K₂P family, whereas Zn²⁺ is a TASK-3 selective blocker relative to TASK-1 and TASK-2 only. Moreover, as the Reviewer indicates, Zn²⁺ also affects a number of other ion (potassium) channels. In the revised manuscript we now use the word “selective” instead of “specific” and discuss in the Results and Discussion Section the limited selectivity of Zn²⁺ (pg 7, lines 23-25; pg 11, lines 23-27).

To address these limitations raised by the reviewer, we have now performed additional experiments with more selective blockers for TASK-1 (ML-365), TASK-3 (PK-THPP), and in addition for TREK-1 (Spadin), now displayed in new Figure 6g-l. We have also additionally investigated the level of expression of members of the K₂P channel family in rabbit atrium by RNA-sequencing (new Figure 5a). The results from these new experiments (now included in the Results section: (Figure 5a) → pg 7, lines 11-16; (Figure 6g-l) → pg 8, first paragraph), indicate that the involvement of either of these 3 channels is unlikely (Discussion section: pg 11, lines 21-24). Unravelling the exact nature of the Sub-P/Senkide-sensitive outward current and its regulatory pathways is outside the scope of the present paper, and constitutes a separate study in its own right.

Response to Reviewer #2

We thank the Reviewer for the stimulating comments and useful suggestions.

In response to the comments of the Reviewers, we have made substantial revisions to the manuscript and have performed additional experiments, including implementation of time controls, measurements with more specific blockers for TASK-1, TASK-3 and TREK-1, additional molecular analyses including RNA-sequencing, and assessment of the anti-arrhythmic action of Senktide in a rabbit isolated heart model of AF.

The Reviewer's comments are included in italic typeface, our response in regular typeface.

I have some concerns about the complete reliance on inhibitors, with little discussion about selectivity (or lack thereof) The authors should be more forthcoming on this issue.

The effectiveness and selectivity of specific ion-channel block was not verified in general. This may be less important for the broad conclusion from Fig 2, but seems missing for Fig 3, where the Ca, NCX and Cl(Ca) currents reported should be shown to be sensitive to the inhibitors used elsewhere (e.g. Cd or nifedipine for Ca current, Ni for NCX and DIDS for Cl(Ca)). This concern is especially critical for the use of 100 μ M Zn as a specific TASK-3 inhibitor (later in Fig 7). That concentration of Zn is expected to influence many things.

The Reviewer's point is well taken. The Reviewer is right that the blockers used for the inhibition of the respective ion channels for the experiments shown in Figure 2, are well-accepted blockers in cardiac electrophysiology. Those used for the experiments shown in Figure 3, however, are less common and the choice for these blockers should have been justified. The currents shown in Figure 3 were measured with specific solutions and voltage protocols in presence of selective blockers to exclude contribution of membrane currents, other than the current type under investigation. In short, I_{CaL} was measured under conditions where only Ca^{2+} is active; I_{NCX} is measured as $NiCl_2$ -sensitive current as described in detail by Hinde et al., 1999; and $I_{Cl(Ca)}$ is measured during depolarizing pulses in presence of I_{Na} and I_{To1} blockade (Zygmunt et al., 1992). In the Methods section of the revised manuscript (pg 17, lines 1-21), we now provide the relevant references for each of these blockers with respect to their selectivity.

As for the use of $ZnCl_2$ as a selective blocker for TASK-3, the Reviewer is correct: Zn^{2+} is a TASK-3 selective blocker in comparison with its effects on TASK-1 and TASK-2, but besides that it also affects other ion channel types. In the Result section (pg 7, lines 23-25) and Discussion section (pg 11, lines 23-27) we now discuss this limited selectivity of Zn^{2+} . Moreover, in view of this limited selectivity, we have performed additional action potential experiments with more selective blockers for TASK-1 (ML-365) and TASK-3 (PK-THPP). In addition, we also included a selective blocker for TREK-1 (Spadin), now displayed in Figure 6g-l.

The results from these new experiments, indicate that the involvement of either of these 3 channels is unlikely (Results section: pg 8, first paragraph; Discussion section: pg 11, lines 21-24). Hence, despite the clear AP prolonging and anti-arrhythmic effects of Senktide (see also below, new Figure 8, and Results section (pg 9, lines 13-end)), the underlying biophysical process remains incompletely understood. Unravelling the exact nature of the SP/Senktide-sensitive outward current is outside the scope of this paper, and constitutes a separate study in its own right.

Another concern is that the links between these novel observations about the role of the NK-3 receptor in atrial APD prolongation and the intrinsic cardiac nervous system (ICNS) as stated in the Abstract and Introduction) are weak here. It seems more prudent for the authors not to muddy the waters by talking so much about the ICNS. This point could be mentioned in the Discussion rather than abstract/intro), but the text should focus on the fact that the NK-3 receptor may be a novel target for treatment/prevention of AF.

Thank you for the suggestion to alter the storyline of the paper. We have followed your advice and have removed any mention of the ICNS from the Abstract and in the Introduction and Discussion only shortly mention the intracardiac ganglia as the source of neurokinin production/release. We agree that the focus should be on the on the fact that the NK-3 receptor may be a novel target for treatment/prevention of AF. Hence, also in response to the comments of Reviewer 3, we have performed a new set of experiments in which we tested and confirmed the anti-arrhythmic properties of NK-3R stimulation in a rabbit isolated heart model of AF (new Figure 8, Results section pg 9, lines 13-end). The findings from these additional experiments further underscore the importance of the anti-arrhythmic potential of NK-3R stimulation.

A related point is that the authors should directly discuss interaction of this substance P effect with other autonomic neurotransmitters that may be locally co-secreted with substance P in the heart. Indeed, a few experimental tests may be worthwhile. Substance-P is released by nerves in the ICNS, but Sub-P has highest affinity for NK-1 receptors (located on neurons and other cell types) and lowest affinity for NK-3 receptors (myocytes). So at physiological Sub-P levels (which may not be known in the heart), the predominant action of Sub-P may be on neurons, where it stimulates cholinergic neurons, and which would likely produce atrial APD shortening, not prolongation. I'm pretty sure this is why Sub-P was used in the manuscript to study isolated atrial myocytes, but a specific NK-3 receptor agonist was used in the whole-heart and in vivo experiments, leaving this question open (maybe that's why they used the selective agonist). Therefore, relating these findings to physiological function of the ICNS is a little bit of a stretch as it stands. This doesn't detract from the importance of the findings, though, with respect to the NK-3 receptor specifically. Just maybe influences how they should tell the story.

The Reviewer raises an important point. Indeed, Sub-P may be co-released with several other neuropeptides (ACh, CGRP, NKA) and stimulates cholinergic neurons to release ACh. We therefore agree with the Reviewer that it is difficult to draw any definite conclusions on the physiological function of Sub-P released from the ICNS on atrial electrophysiology based on our present findings. Sub-P was the starting point of our study for reasons indicated in the Introduction. However, in view of the subsequent finding that its AP prolonging effect is entirely mediated through the cardiac NK-3 receptor, we propose that the endogenous agonist neurokinin B (which has a much higher affinity for NK-3R), has more relevance in terms of physiological function. The latter consideration has now been included in the Discussion section (pg 10, lines 27-33).

Other points:

Page 4. The 6 mV depolarization of resting Em with Sub-P is surprising and begs the question as to how much IK1 vs. Sub-P sensitive K current controls diastolic Em and stability with respect to triggered arrhythmias like DADs in atrium (vs. ventricle). It is also a bit surprising that the 6 mV depolarization did not reduce Na channel availability significantly (as in Vmax) and conduction. Pg 4 Line n-4 " a background K-selective

channel current.”

While Sub-P indeed caused a RMP depolarization of 6 mV, this is likely not due to the NK-3R effect of Sub-P, as selective NK-3R stimulation by Senktide does not depolarize the RMP (Figure 4, table 1). Moreover, the inward component of the Senktide-sensitive current is virtually absent, whereas it is clearly present for the Sub-P sensitive current (Figures 6 and 2 resp.). We propose that this difference can be explained by the assumption that Sub-P in cardiomyocytes also inhibits the inward rectifier current (as shown in neurons) and does so e.g. by activating the NK-1 and/or NK-2 receptor in addition to the NK-3 receptor. (Shen K.Z. and North, R.A. Substance P opens cation channels and closes potassium channels in rat locus coeruleus neurons. 50, 345–353 (1992)).

We agree with the Reviewer that the observed 6 mV depolarization of RMP would be expected to affect V_{max} . However, while a decrease in V_{max} was indeed observed in the majority of cells, this did not reach statistical significance due to the large variation. The latter was already mentioned in the original manuscript, but in the revised manuscript, we have reworded the Results section related to this issue (page 4, lines 11-13).

Pg 7, last full paragraph. Are you implying that Sub-P is working via Gs-alpha here? Via cAMP and PKA? Or is the NK-3 receptor coupled to TASK-1 & 3 via Gq11 as mentioned in the Discussion? Try to be clearer about what is known/expected and hypothesized.

We apologize for the confusion. Indeed, we imply that neurokinin receptors are coupled to the Gq/11 subgroup of G-proteins, not Gs-alpha. To clarify this, we have reworded the Results section related to this issue (pg 7, lines 7-11) and now use consistently the terminology Gq/11. We have now also indicated more clearly what is known, e.g. “has been reported”, and what is hypothesized, e.g. “we considered likely candidates”.

Fig 8. Is there any immunofluorescence control data to show that the NK3R antibody is specific and not picking up some other protein (a notorious problem for many antibodies –especially for sparsely expressed channels and receptors)? Even showing parallel ventricular myocyte data as a negative control would be somewhat reassuring.

To address this issue, we performed additional immunohistochemistry analyses. Since the antibody used in the original manuscript was unfortunately no longer available at the manufacturer, we repeated stainings in both rabbit and human atrial tissue using different anti-NK-3R antibodies (Rabbit: Abcam, ab123303; Human: Immunostar 20061). These new stainings showed similar results as the original immunohistochemistry data (see new Figure 5b in the revised manuscript), demonstrating NK-3R labelling on the membrane of rabbit and human atrial myocytes. No significant staining (other than a weak background signal) was observed in negative controls (i.e. secondary antibody only). Moreover, specific labelling for NK-3R was absent from rabbit spleen tissue, which has been shown to be devoid of NK-3R [Pinto F.M., Almeida T.A., Hernandez M., Devillier P, Advenier C., Candenas M.L. mRNA expression of tachykinins and tachykinin receptors in different human tissues. Eur J Pharmacol. 494, 233–239 (2004). doi:10.1016/j.ejphar.2004.05.016]. These results are now included in the Supplement (Supplementary Figure 4) of the revised manuscript.

Discussion, pg. 19: “On the contrary, a decrease in inward current would lengthen the AP.” I think you mean shorten the AP....?

Figure 7 was missing, but I got it before submitting this review.

We thank the Reviewer for pointing out this mistake. Indeed we erroneously wrote “lengthen”, but this should of course have been ”shorten”. This has now been corrected accordingly.

We apologize for the absence of Figure 7.

Figure 7 was present in the original manuscript, but was somehow ‘lost’ during conversion to a PDF file in the process of submission to Nat. Comm.

Response to Reviewer #3

We thank the Reviewer for the stimulating and favourable comments and useful suggestions.

In response to the comments of the Reviewers, we have made substantial revisions to the manuscript and have performed additional experiments, including implementation of time controls, measurements with more specific blockers for TASK-1, TASK-3 and TREK-1, additional molecular analyses including RNA-sequencing, and assessment of the anti-arrhythmic action of Senktide in a rabbit isolated heart model of AF.

The Reviewer's comments are included in italic typeface, our response in regular typeface.

(Patho-)physiology. What I really miss is evidence for the involvement of Sub-P in atrial (patho-)physiology. This could relatively easily be achieved with the Langendorff set-up of isolated rabbit hearts. By tachypacing, the vulnerability to AF could be evaluated. This should be done both in the presence of agonists and antagonists. Langendorff heart perfused with saline buffer will be slightly ischemic and as this might trigger Sub-P release application of an antagonist would be of interest. Also, application of Sub-P, and not only Senktide should be done. Another option is to use the thoracotomised rabbit model for these investigations.

We thank the Reviewer for this suggestion. Sub-P was the starting point of our study as a number of previous studies suggested a protective effect of Sub-P in AF. However, in view of our subsequent finding that its AP prolonging effect is entirely mediated through the NK-3 receptor, we considered the endogenous agonist neurokinin B (which has a much higher affinity for NK-3R) to be more relevant in terms of (patho)physiological function. We therefore performed additional experiments to assess the anti-arrhythmic potential of Senktide (an NKB-analogue) in a rabbit isolated heart model of AF based on atrial dilatation (new Figure 8; Results section: pg 9, lines 13-end; Discussion section: pg 12, lines 3-8). We found that the increase in atrial effective refractory period (AERP) by NK-3R stimulation strongly suppressed duration and incidence of AF. These observations underscore the importance of the NK-3R as a pharmacological target to prevent and terminate AF.

Power. The number of experiments performed in most of the procedures are very low. In particular the n numbers in the ex vivo experiments are critically low. Please include power calculations/arguments or/increase numbers.

We appreciate the concern of the Reviewer towards the low n numbers. We would first like to stress that all experiments in this study were performed in a paired fashion, giving rise to more power than for unpaired experiments, and thus allowing for a lower n number (using a default power of 0.8). Power calculations for the cellular electrophysiology experiments yield powers around 0.8 (e.g. Sub-P 1 μ M APD₉₀: n=5, avg0=105, avg1=129, sigma=17, power=0.88; Sub-P 10 μ M steady-state outward current: n=5, avg0=4.1, avg1=3.0, sigma=0.9, power=0.78; Senktide 20 nM APD₉₀: n=9, avg0=107, avg1=147, sigma=38, power=0.89; Senktide 50 nM steady-state outward current: n=8, avg0=6.1, avg1=4.7, sigma=1.3, power = 0.87).

(http://www.statisticalsolutions.net/pssZtest_calc.php). The Reviewer's main concern related especially to the *ex vivo* experiments, presumably referring to the experiments of original Figure 6A,C in which we determined AERP in the rabbit Langendorff heart, and in human LAA tissue. In the additional experiments performed in the isolated heart model of AF (see above), we again determined the AERP (n=7). The power for this new set of AERPs is 0.90 (n=7, sigma=17). As for the human LAA tissue: unfortunately, the availability of the tissue is limited and in the time-

span of the revision, we only had the opportunity to increase the n number by 1 from 4 to 5 (new Figure 8b). Nevertheless, in our view these experiments are valuable, since they indicate that a similar mechanism (APD regulation by NK-3R stimulation) is operative in human atrium.

Concentration of Senktide. For the rabbit Langendorff experiments you have used 20-50 nmol/l of Senktide. I don't understand this. Why not the same concentration? And if not, can you at all group these results?

We agree with the Reviewer that this was not optimal. Hence, in the revised version of the manuscript we have repeated these AERP measurements in the rabbit Langendorff AF model at the single dose of 20 nM (see also response to previous comment). These additional data are presented in Figure 8a and the Results section (pg 8, last paragraph).

Time-matched controls. It seems like the electrophysiological experiments are performed in a paired fashion with control conditions followed by drug application. This is an acceptable way to perform the experiments, however, it is necessary to confirm that time from break-through/explantation is not a factor influencing the recordings. This has not been done.

Thank you for raising this important issue. Indeed, all experiments were performed in a paired fashion. To ascertain that action potential prolongation is not due to a time effect, we have now performed time-matched controls for the cellular experiments, as suggested by the Reviewer. These new data are presented in Supplementary Figure 3 of the revised manuscript, and mentioned in the Results Section (pg 6, lines 26-29). We did not consider it necessary to perform time controls also for the Langendorff-perfused hearts, as in these experiments the Senktide effects were reversible upon wash out.

Immunohistochemistry. The stainings reveal a nice membrane-like expression of NK-3R in rabbit and human atrial myocytes. The primary AB is used in a 1:20 dilution. This reviewer miss the controls. Please refer to other publications where this AB has been carefully evaluated, or conduct/show this experiments yourself.

To address this issue, we performed additional immunohistochemistry analyses. Since the antibody used in the original manuscript was unfortunately no longer available at the manufacturer, we repeated stainings in both rabbit and human atrial tissue using different anti-NK-3R antibodies (Rabbit: Abcam, ab123303; Human: Immunostar 20061). These new stainings showed similar results as the original immunohistochemistry data (see new Figure 5b in the revised manuscript), demonstrating NK-3R labelling on the membrane of rabbit and human atrial myocytes. No significant staining (other than a weak background signal) was observed in negative controls (i.e. secondary antibody only). Moreover, specific labelling for NK-3R was absent from rabbit spleen tissue, which has been shown to be devoid of NK-3R [Pinto F.M., Almeida T.A., Hernandez M., Devillier P., Advenier C., Candenas M.L. mRNA expression of tachykinins and tachykinin receptors in different human tissues. *Eur J Pharmacol.* 494, 233–239 (2004). doi:10.1016/j.ejphar.2004.05.016]. These results are now included in the Supplement (Supplementary Figure 4) of the revised manuscript.

Minor:

Line 77-83 could be condensed as they are more or less repeating information from abstract and other places in the MS.

We have reworded the abstract and the last part of the Introduction, however, as it is required for Nat. Comm. to include a last paragraph in the Introduction section, containing a brief summary of the results and the conclusions, some overlap is unavoidable.

Line 107-108: At which time-point after the step was this recorded?

Steady-state currents were measured as the current at the end of the 500-ms voltage step. This information is now included in the Methods section (pg 16, last line, pg 17, first line).

Line 225. Has any compound ever been found to be specific? And a divalent ion? You should delete “specific”.

The Reviewer is completely right, we have erroneously used the terminology ‘specific’. In the entire manuscript we have now replaced ‘specific’ by ‘selective’.

Figure 1: You should show RMP and APA graphs. These are important when trying to understand the effect on AF vulnerability etc.

We thank the Reviewer for this suggestion. The RMP and APA values are now not only displayed in the Table 1 (original MS), but also in Figure 1 of the revised manuscript.

Figure 1-7: Time-matched controls – at least in some of the figures

For answer please see earlier response on comment time-matched controls.

Figure 6: Why do you use Senktide instead of Sup-P? This should be explained more carefully in the text.

Sub-P was the starting point of our study as a number of previous studies suggested a protective effect of Sub-P in AF. However, in view of our subsequent finding that its AP prolonging effect is entirely mediated through the NK-3 receptor, we considered the endogenous agonist neurokinin B (which has a much higher affinity for NK-3R) to be more relevant in terms of (patho)physiological function. We therefore performed the AERP measurements (old Figure 6, new Figure 8a,b), and additional experiments to assess the anti-arrhythmic potential of Senktide (an NKB-analogue) in a rabbit isolated heart model of AF based on atrial dilatation (new Figure 8c,d; Results section: pg 9, lines 13-end; Discussion section: pg 12, first paragraph). An explanation for the use of Senktide stating: “As the AP prolonging effect of Sub-P is entirely mediated through the NK-3R, we maintained the use of Senktide - a potent analogue of the endogenous agonist of the NK-3R (neurokinin B) - in all following experiments” (Results section: pg 6, last line; pg 7, lines 1-2).

REVIEWERS' COMMENTS:

Reviewer #1 (Remarks to the Author):

Thank you for the revised manuscript. Albeit I agree that the manuscript has strongly improved during the revision, I stick to my initial evaluation that the manuscript falls mechanistically too short and that the nature of the current and the patch clamp recordings are in parts too weak or error prone. Although the authors now more carefully use or interpret the data with unspecific drugs, the currents they isolated are obscure (Figure 6C) and variable (2C, 6C, 6F). The difference current in Figure 6F has a reversal potential of -50 mV and is thus clearly not a background potassium current as the authors claim. It also seems to have a threshold around -50 mV which would rather fit to a voltage-activated (Kv) channel. Also the other difference currents do not match in shape with one of the K2P channels described in the literature. As the nature of the current is unclear and the difference currents are "odd", it is essential to identify what these conductances are or whether they are artificial. Within the revision the authors could rule out TREK-1, TASK-1 and TASK-3 by pharmacological means which is however a strange experimental design and I wonder why they did not study KCNK5 (TASK-2) and KCNK6 (TWIK-2) which showed the highest expression levels besides KCNK3 (TASK-1) in the tissue they examined. In addition, do these channels show an atrium-specific expression? Only under these circumstances these two could contribute to the effects. The other K2P channels are ruled out by their experiments (patch or PCR) and given the somewhat strange difference currents there are doubts that it is really a K2P or background channel that is described here. In addition the authors describe that they have effects on ERP in human cardiomyocytes, but there are none, as the effects are not significant. These effects are not statistically significant, $p=0.06$, thus one cannot present figures with typical or representative changes if there are no significant changes. Also in the Discussion section the authors raise the point that in human atrial cardiomyocytes the ERP is changed (which it is not!). In this context it might be worth looking why only rabbits have an altered ERP. Is there a K2P channel present in rabbit that is not expressed in humans? In this context, in the Discussion section in which the authors state that K2P channels can alter AP duration, I think that there are not suitable references provided. The first experimental data provided to prove that K2P channels can alter cardiac action potential duration were by Putzke et al., 2007 for rats and Limberg et al., 2011 for humans. Minor comments are "CsCl₂" which is often used in the Methods section. Most importantly, if ERP is not significantly changed in humans, how can the authors claim that it is a good target against atrial fibrillation!?

Reviewer #2 (Remarks to the Author):

The authors have been highly responsive and the ms is significantly improved.

Reviewer #3 (Remarks to the Author):

The authors answered all my questions sufficiently. I have no further questions.

Response to Reviewer #1

We thank the Reviewer for the valuable comments.

The Reviewer's comments are included in italic typeface, our response in regular typeface.

... Although the authors now more carefully use or interpret the data with unspecific drugs, the currents they isolated are obscure (Figure 6C) and variable (2C, 6C, 6F). The difference current in Figure 6F has a reversal potential of -50 mV and is thus clearly not a background potassium current as the authors claim. It also seems to have threshold around -50 mV which would rather fit to a voltage-activated (Kv) channel. Also the other difference currents do not match in shape with one of the K2P channels described in the literature. As the nature of the current is unclear and the difference currents are "odd", it is essential to identify what these conductances are or whether they are artificial.

The Sub-P-sensitive currents and the Senktide-sensitive current in Figures 2C and 6C resp., both reverse sign between -70 mV and -80 mV, close to the reversal potential for K⁺-ions. The ZnCl₂-sensitive current indeed reverses sign at somewhat less negative voltages between -50 mV and -60 mV. However, as pointed out by this reviewer at an earlier instance, ZnCl₂ is not a very specific blocker of TASK-3 (TREK-1, TASK-1) channels, but may also affect ion channels/exchangers carrying other ion currents than potassium currents, potentially resulting in a deviation of the K⁺ reversal potential of the ZnCl₂-sensitive current.

The shape of the I-V relation does not necessarily implicate a threshold of -50 mV, but can equally well be explained by (strong) outward rectification.

A large part of the biophysical characterizations of K2P channels in the existing literature have been performed in (artificial) expression systems (e.g. HEK-cells, oocytes) in which one can obtain robust currents. In these systems, interactions with other membrane-proteins that may affect channel behaviour are limited or absent, and one can selectively study one particular ion-channel type. The background K⁺-currents we describe in our study were measured as difference currents (Sub-P-sensitive currents or Senktide-sensitive currents) in their natural environment in freshly isolated cardiomyocytes. For this reason it is to be expected that their biophysical characteristics are not completely identical to those in expression systems and that the shape of the I-V curve is not an exact match. The natural environment is the preferred condition to study these currents in this case.

In this respect, several reports in literature on K2P currents measured in isolated atrial cardiomyocytes have shown reversal potentials of approx. -50/-60 mV, or even -30 mV, (Limberg et al. Cell Physiol Biochem. 28, 613-624 (2011); Schmidt et al. Circulation. 132, 82-92 (2015); Harleton et al. AJP: Heart and Circulatory Physiology, 308(2), H126-34. (2015))

We agree that the nature of the K⁺ background current is still unclear and that it is important to resolve the underlying channel type. Nevertheless, as was indicated in the Discussion section, we feel that this is beyond the scope of the present study and that it requires a separate study fully dedicated to exact channel identification.

We therefore respectfully disagree with the reviewer that the currents are obscure or odd.

Within the revision the authors could rule out TREK-1, TASK-1 and TASK-3 by pharmacological means which is however a strange experimental design and I wonder why they did not study

KCNK5 (TASK-2) and KCNK6 (TWIK-2) which showed the highest expression levels besides KCNK3 (TASK-1) in the tissue they examined.

We did not study *KCNK5* and *KCNK6* because the RNA sequencing experiments showing high expression levels of these channels, were performed in the revision stage of the manuscript. In the initial study we investigated TASK-1 and TASK-3 channels, as we considered these likely candidates, justified by their known functional presence in (human) atrial myocytes and their inhibition by Gq-proteins. As stated before, we felt that further exact channel identification deserved a separate study fully dedicated to this purpose.

In addition, do these channels show an atrium-specific expression? Only under these circumstances these two could contribute to the effects.

We disagree with the reviewer that only under conditions of atrial-specific expression of these K2P channels, these channels can contribute to the NK-3 receptor mediated prolongation of the atrial action potential. The AP prolongation by NK-3 receptor stimulation can be atrial-specific (i.e. absent from ventricle) by two mechanisms: either the K⁺ channels inhibited by NK-3 receptor stimulation are expressed specifically in the atrium (as suggested by the reviewer), or the NK-3 receptor is specifically expressed in the atrium, in which case the absence of the K2P channels from the ventricle is not a required to accomplish an atrial-specific effect.

..... the authors describe that they have effects on ERP in human cardiomyocytes, but there are none, as the effects are not significant. These effects are not statistically significant, p=0.06, thus one cannot present figures with typical or representative changes if there are no significant changes. Also in the Discussion section the authors raise the point that in human atrial cardiomyocytes the ERP is changed (which it is not!). In this context it might be worth looking why only rabbits have an altered ERP. Is there a K2P channel present in rabbit that is not expressed in humans? Most importantly, if ERP is not significantly changed in humans, how can the authors claim that it is a good target against atrial fibrillation!?!.....

The reviewer is (technically) right that the ERP-prolonging effect of Senktide in human LAA did just not reach statistical significance (p=0.06), when tested in a paired fashion. However, this does not mean that the observed effect is not significant in the sense of important. Prolongation of the ERP occurred in all 5 human LAA's tested and it is reasonable to assume that statistical significance (p<0.05) was just not achieved due to the small sample size in combination with the large variation in effect size. In this case we would have preferred to use a non-parametric test. However, this required a minimum sample size of n=7. We should like to point out that the availability of human atrial appendages from patients with AF is rare and that observations on these human tissues are of highly added value.

As we consider the observed effects in human LAA of importance, we would like to maintain this statement. We have now more carefully rephrased the concerning text.

- In the abstract we have removed "atrial myocytes" and "atrial appendages" from the following 2 sentences:

.... We here demonstrate that the neurokinin substance-P (Sub-P) activates a neurokinin-3 receptor (NK-3R) in rabbit **and human atrial myocytes**, prolonging action potential (AP) duration

.... NK-3R stimulation lengthened atrial repolarization in intact rabbit hearts and in **human atrial appendages**....

Instead we added a separate sentence stating: “In human atrial appendages, the phenomenon of NK-3R mediated lengthening of atrial repolarization was also observed”.

- In the Results section (pg 9, line 9-12): We removed the word “typical” and included the following sentence: “Prolongation of the ERP occurred in all 5 human LAA’s tested”

- In the Discussion section (pg 10, line 5): We removed “as well as in human atrial appendages” and included the following sentence “Also in human atrial appendages the latter phenomenon was observed”.

In this context, in the Discussion section in which the authors state that K2P channels can alter AP duration, I think that there are not suitable references provided. The first experimental data provided to prove that K2P channels can alter cardiac action potential duration were by Putzke et al., 2007 for rats and Limberg et al., 2011 for humans.

We think the reviewer is mistaken. In the Discussion section we refer to studies that demonstrate that K2P channels are involved in repolarization of the **atrial** action potential, not in cardiac repolarization in general. For that we refer to 3 studies (ref 26-28, page 22 of the Discussion section), among which Limberg et al. 2011. We did not refer to the study of Putzke et al. as that study was performed in rat **ventricular** myocytes.

Minor comments are “CsCl₂” which is often used in the Methods section.

We thank the reviewer for being observant. We now replaced ‘CsCl₂’ by ‘CsCl’ at the 2 occasions in manuscript.